# Fast photothermal spatial light modulation for quantitative phase imaging at the nanoscale

Hadrien M. L. Robert [1], Kristýna Holanová[1], Łukasz Bujak [1], Milan Vala [1], Verena Henrichs [2], Zdeněk Lánský [2] & Marek Piliarik [1✉]

Spatial light modulators have become an essential tool for advanced microscopy, enabling breakthroughs in 3D, phase, and super-resolution imaging. However, continuous spatial-light modulation that is capable of capturing sub-millisecond microscopic motion without diffraction artifacts and polarization dependence is challenging. Here we present a photothermal spatial light modulator (PT-SLM) enabling fast phase imaging for nanoscopic 3D reconstruction. The PT-SLM can generate a step-like wavefront change, free of diffraction artifacts, with a high transmittance and a modulation efficiency independent of light polarization. We achieve a phase-shift > π and a response time as short as 70 μs with a theoretical limit in the sub microsecond range. We used the PT-SLM to perform quantitative phase imaging of sub-diffractional species to decipher the 3D nanoscopic displacement of microtubules and study the trajectory of a diffusive microtubule-associated protein, providing insights into the mechanism of protein navigation through a complex microtubule network.

[1] Institute of Photonics and Electronics of the Czech Academy of Sciences, Prague 18251, Czech Republic. [2] Institute of Biotechnology of the Czech Academy of Sciences, BIOCEV, Vestec, Prague West 25250, Czech Republic. ✉email: piliarik@ufe.cz

nterferometric microscopes take advantage of the interference between at least two light waves, usually a reference and a probe wave, to get information about microscopic objects. Numerous methods based on this principle have been developed, including the phase-contrast microscope[1], the holographic microscope[2], the Nomarski microscope[3], and the interferometric reflection microscope[4]. In addition, by shaping the reference wave, the phase profile of the probe wavefront can be reconstructed[5]. This approach, established as quantitative phase imaging (QPI), has become a powerful tool for capturing phase images of transparent samples in a transmission configuration, including high-resolution 3D profiling of single cells[6] and mapping the phase of opaque structures in a reflection configuration[7]. QPI has already reached the sensitivity to image the displacement of nanostructures at video-rates[8,9] and specific applications in plasmonic imaging have allowed characterization of nanoscopic patterns[10,11]. However higher sensitivity and speed of QPI remain limited by the performance of the available spatial light modulation techniques. Recently, interferometric scattering microscopy (iSCAT) has become the method of choice for ultrasensitive imaging and localization of sub-wavelength scatterers including unlabeled single proteins with a high spatiotemporal resolution[12–14]. In the iSCAT microscope, the light wave scattered on the sample and the reference wave reflected from the sample substrate share a simple imaging path and interfere on a detector. However, any direct control of the phase-shift in iSCAT microscopy is challenging owing to its strong sensitivity to small wavefront distortion[15].

There are several modern techniques for wavefront shaping including deformable mirrors, micromirror devices, or liquid crystal spatial light modulators (LC-SLMs)[16–18]. LC-SLMs have become the central tool for beam shaping in microscopy featuring high resolution (few micrometers) and high definition (several millions of pixels) at affordable prices. These tools have found numerous groundbreaking applications such as point-spread-function engineering for 3D fluorescence localization[19], imaging through scattering media[20], and stimulated emission depletion (STED) microscopy[21]. However, owing to their intrinsic properties, wave-shaping using LC-SLMs has never been successfully implemented in an ultrasensitive iSCAT setup. LC-SLMs have limited phase stability, are polarization-sensitive, suffer from strong diffractive effects, and have an intrinsic response time of several milliseconds. To enhance the phase modulation speed to the kHz range, technologies such as ferroelectric LC-SLMs and micromirror devices are used at the cost of the diffraction efficiency and the possibility of only discrete (usually on-off) modulation[17].

To overcome these limitations, spatial-light modulators based on thermo-optic effects have been recently introduced. Thermo-optic modulation is well established in guided wave optics via plasmonic modulators[22,23] and with active metasurfaces for infrared wavelengths[24,25]. It has been employed in combination with thermoplasmonic-base temperature control[26] to perform photothermal imaging[27], measure temperature gradients[28,29] and shaping temperature profiles by nanoparticle distribution[30] or spatial modulation of a light wave[31,32]. The photothermal control enabled all-optical light modulation based on local tuning of the birefringence of liquid crystals[33], the generation of adjustable thermal lenses[34,35], or the implementation of thermally driven wavefront shaping with multiple photothermal lenses[36,37]. However, the speed and lateral resolution of photothermal wavefront shaping remains limited by the confinement volume of the temperature change.

Here, we introduce a photothermal spatial light modulators (PT-SLM) enabling the adjustment of the phase-shift between the scattered light and the reference beam in iSCAT microscopy. We implement the technology for quantitative phase imaging by placing the PT-SLM in a Fourier plane of an iSCAT microscope and extract the full phase information of the light scattered on weak scatterers. We characterize the 3D phase profiles of light scattered on sub-wavelength scatterers, demonstrate the 3D reconstruction of biological nanostructures at the kHz rate, and show that such high-speed 3D tracking of single proteins reveals mechanics of biological systems often hidden in time-averaged data.

## Results

**PT-SLM principle and numerical simulations.** The thermo-optic effect is used to phase-shift an optical wave propagating through a thermo-optic material, whose refractive index is highly sensitive to temperature change. The usual concept of thermo-optic phase modulation is based on the semi-infinite heat propagation in the thermo-optic material around a heat source located on an adjacent substrate[34,35]. By this means, a thermal lens of a polynomial profile and an arbitrary mean diameter can be generated; however, the spatial resolution and the response time is intrinsically limited by the dimension of the thermal lens. To solve the limitation of the heat propagation, we introduce the concept of confining the generated temperature change within the desired area by optimizing the heat gradients and dissipation rates in a three-layer design, as illustrated in Fig. 1a. Incident plane waves illuminate a layered structure made of a glass BK7 substrate, a thin layer of thermo-optic material, i.e., liquid glycerol, and a transparent heat sink superstrate made of sapphire. To trigger the heat distribution on the boundary of the thermo-optic material, we use the photothermal effect in a layer of gold nanorods patterned on the glass substrate[26]. In essence, the heat generated on one side of the thermo-optic material builds a laterally uniform temperature gradient across the layer of the thermo-optic material and dissipates into the transparent heat sink. The temperature change within the thermo-optic material varies its refractive index, resulting in phase retardation of the transmitted optical wave.

To study the effect of a microscopic heat source on the optical phase of incoming light, we used a theoretical model solving the static heat equation in a three-layered system[38] (detailed in Supplementary Note 1). The geometry of the numerical model is shown in Fig. 1a. We modelled a heat source of a fixed diameter of 60 µm delivering a heating power of $Q = 4.8 \, \mu\text{W}\mu\text{m}^{-2}$ interfaced with a 20-µm-thick layer of thermo-optic material (glycerol considered in the model). The 3D temperature profile induced by the heat source in a steady-state is depicted in Fig. 1b–c, and the resulting 3D distribution of the refractive index is shown in Fig. 1d–e. We observed that the refractive index gradient is confined within the thermo-optic material. This confinement is owing to the thermo-optic coefficient, $\triangle n / \triangle T$, of glycerol, which exceeds that of glass by a factor of >10. The resulting phase-shift cross-section of normally incident plane waves ($\lambda = 488$ nm) transmitted through the structure is shown in Fig. 1f. In the model, the phase-shift peaks at $0.9\pi$ in the center of the heated area and drops rapidly outside the heat source. The strong thermal gradient around the heat source is explained by the thermal conductivity of the superstrate, which is >100-times higher for sapphire than for glycerol (details in Supplementary Fig. 2). The effect of the thickness of the thermo-optic material is shown in Fig. 1g. The green curve (corresponding to the largest thickness of the thermo-optic material) illustrates the phase profile of a thermal lens, which is similar to that generated through a semi-infinite configuration of thermo-optic wavefront modulation[34,35]. Clearly, decreasing the thickness results in a phase-shift profile more confined to the area of the heat source,

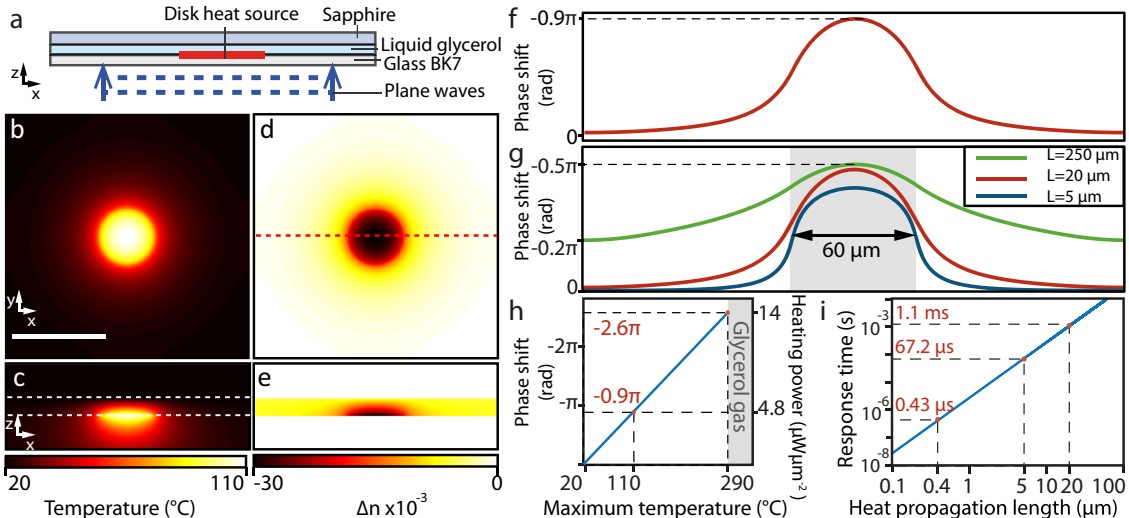

**Fig. 1 Numerical simulation of the spatial phase modulator. a** Illustration of the structure used. **b** 2D map of temperature in the *xy* cross-section and (**c**) *xz* cross-section (scale bar indicates 100 μm). The boundaries of the thermo-optic material are indicated by the two white dashed lines. **d** Resulting refractive index variation in the *xy* cross-section, and (**e**) in the *xz* cross-section. **f** Phase-shift profile of a plane wave propagating through the structure in *z* at the position of the red dashed line in (**d**). **g** Phase-shift profiles for different thicknesses of the thermo-optic layer. Heating powers of 0.4 μWμm⁻², 2.6 μWμm⁻², and 14 μWμm⁻² were considered for thicknesses of 250 μm, 20 μm, and 5 μm, respectively. **h** Peak phase-shift and corresponding heating power as a function of the peak temperature change (the glycerol thickness is 20 μm). **i** Response time of the temperature change as a function of the characteristic dimension according to the model.

and a more uniform phase-shift profile within the heated area. Indeed, the lateral distribution of the phase shift around the edge of a heat source is limited by the temperature gradient generated in the layer of thermo-optic material, and consequently by the thickness of the layer. The peak phase-shift scales linearly with the maximum temperature until the boiling temperature of glycerol is reached, as shown in Fig. 1h, which corresponds to a maximum phase-shift of 2.6π. As shown in Supplementary Fig. 3, this maximum can be increased by increasing the thickness of the glycerol layer.

The characteristic response time $\tau$ of a thermal process to be in a steady-state is estimated from the heat equation and is expressed as,

$$\tau = \frac{L^2}{4D}, \qquad (1)$$

in which $L$ is a characteristic size of the system, e.g., the thermo-optic layer thickness, or the heated area diameter depending on which parameter limits the heat propagation, and $D$ is the thermal diffusivity of the medium[39]. Considering the glycerol thickness of $L = 20$ μm and the thermal diffusivity of $D = 0.096$ mm²s⁻¹, the response time yields $\tau = 1.1$ ms. The dependence of the response time is shown in Fig. 1i. The reduction of the glycerol layer thickness to 5 μm decreases the response time to 67.2 μs. We estimate a minimum achievable response time of 430 ns corresponding to the smallest optically addressable area, which is limited by the diffraction limit (~400 nm).

**Implementation of the PT-SLM.** We prototyped a PT-SLM and combined it with an iSCAT microscope. To generate the disk-shaped heat source, we used the photothermal effect of a layer of gold nanorods. Gold nanorods with an average size of 20 nm × 50 nm were immobilized on a glass coverslip to obtain homogeneous spatial distribution and random orientation. The average surface density of the gold nanorods was 300 μm⁻² with <20% variation (Supplementary Fig. 4a). A 20-μm-thick layer of liquid glycerol was sandwiched between the glass coverslip and a sapphire window with the layer of gold nanorods at the glass/glycerol interface. A

continuous laser light source ($\lambda = 660$ nm), providing a heating beam modulated with an acousto-optic modulator, was focused on the surface of the nanorods into a spot of uniform illumination of 60 μm in diameter (inset in Fig. 2a). The surface density of nanorods is sufficiently high to be considered as a homogeneous heat source equivalent to the area of the heating beam[40]. The extinction rate of the layer of nanorods was 13.6% at 660 nm (Supplementary Fig. 4b). An incoming light power of 100 mW within an illuminated surface with a diameter of 60 μm was used to generate a heating power of 4.8 μWμm⁻². Experimental characterization of the spatial phase profile is detailed in Supplementary Fig. 5. Consistently with the theoretical model, the spatial resolution is limited by the isotropic heat propagation and the spatial gradient between the on and off region of the phase shift, which approximately corresponds to the thickness of the glycerol layer.

A schematic diagram of the iSCAT setup is shown in Fig. 2a. A test sample of 30-nm gold nanospheres spin-coated on a glass coverslip is illuminated by a collimated imaging beam (continuous laser, $\lambda = 488$ nm) through a microscope objective. The imaging beam is partially reflected at the coverslip surface and partially scattered on the sample. Both scattered and reflected beams are collected by the microscope objective and routed via a beam splitter through a 4 f system with the thermo-optic modulator system at the common focus to be imaged on a CMOS camera.

The scattered wave and the reference wave interfere on the camera, resulting in the detected intensity of light

$$I_d = I_{inc}\left(r^2 + s^2 + 2rs\cos(\triangle\varphi)\right), \qquad (2)$$

in which $I_{inc}$ is the light intensity incident on the sample, $r$ is the amplitude reflectivity of the imaged interface, $s$ is the scattering amplitude, and $\triangle\varphi$ is the phase difference between the scattered and reference waves. As $s^2 \ll r^2$, the contrast $C$ of the interference image is expressed as[12]:

$$C = 2(s/r)\cos\triangle\varphi. \qquad (3)$$

Considering a nanoparticle at a distance $z$ from the coverslip and surrounded by a medium with a refractive index $n_m$, the term

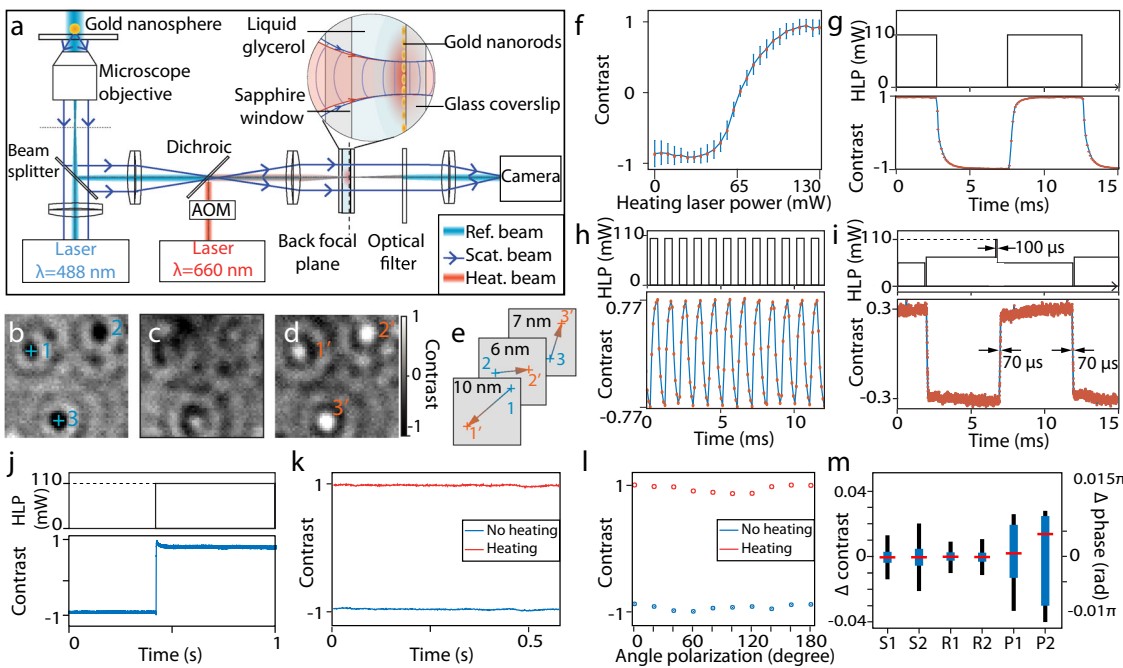

**Fig. 2 Experimental characterization of the PT-SLM. a** Layout of the setup (not to scale, details of the phase modulator structure are magnified). AOM = Acousto-Optic Modulator. **b–d** Normalized iSCAT images of 30-nm single gold nanospheres at heating laser powers $P_{heat}$ (total power incident on the modulator structure) of (**b**) 0 mW, (**c**) 65 mW, (**d**) 110 mW. (Representative images from 21 independent experiments, scale bar indicates 500 nm). Blue and orange crosses and numbers indicate the positions used to assess the lateral distortion. **e** Change in the fitted position of the three nanoparticles indicated in (**b**) and (**d**). **f** Calibration curve of the particle contrast dependence on the heating power. Data are presented as mean values ± SD for $n = 21$ experiments. **g** Time series of the contrast of a gold nanosphere modulated with a rectangular signal at 100 Hz (*HLP* Heating Laser Power) and (**h**) at 1 kHz. **i** Contrast time series for an arbitrary-shaped heating modulation. The upper diagram shows the corresponding temporal profile of the heating beam intensity. **j** Stability of the nanoparticle contrast at a step heat change. **k** Reproducibility of the nanoparticle contrast at a 500 Hz squared modulation—time series plots of on/off equilibrium levels. **l** Polarization dependence of the nanoparticle contrast—time series plots of on/off equilibrium levels. **m** Box-plot of the S1 (off) S2 (on) stability derived from (**j**) ($n = 10{,}000$ images in one experiment), R1 (off), R2 (on) reproducibility derived from (**k**) ($n = 500$ off-on cycles in one experiment), and polarization dependence P1 (off), P2 (on) derived from (l) ($n = 20$ different polarizations in one experiment); the interquartile range is in blue, max–min range in black, and the median in red[53].

$\triangle\varphi$ is defined as $\triangle\varphi = \frac{4\pi}{\lambda} n_m z + \varphi_{Gouy} + \varphi_{scat}$, in which $\varphi_{Gouy}$ is the Gouy phase due to the focus position and $\varphi_{scat}$ is the phase of the polarizability of the nanoparticle. Therefore, the image contrast is determined by the nanoparticle distance from the surface and the Gouy phase due to a change in the focus position. Thus, for a static nanoparticle, a change in the phase of the reference beam results in a contrast variation of the interferometric image. To adjust the phase of the reference beam, we spatially overlapped the Gaussian profile of the reference beam (approximate diameter at half maximum of 30 μm) with the heating beam in the central region of the thermo-optic modulator. The scattered beam propagates through the thermo-optic modulator approximately collimated with a beam diameter of 1.2 mm. The fraction of the scattered beam influenced by the heating beam is as small as 0.25%, which can be considered as negligible compared with a typical signal-to-noise ratio of a single scatterer image, i.e., ~40 for 20-nm gold nanospheres[41]. The measured transmittance of the PT-SLM was 82% at wavelength of 488 nm.

A typical image of nanospheres (diameter = 30 nm) in the interferometric microscope without additional phase-shift modulation is shown Fig. 2b. The image contrast was first set by fine adjustment of the focal position without applying any phase modulation. The effect of increasing the heating laser power to 60 mW and 110 mW is shown in Fig. 2c, d. The dependence of the contrast of the nanospheres image on the light intensity of the heating beam is shown in Fig. 2f (raw data are shown in

Supplementary Fig. 6). To change the phase delay between the interfering beams by π, an absolute change in the light intensity of the heating beam by 70 mW was required, which is equivalent to a delivered heat of $\sim Q = 3.4\,\mu W\mu m^{-2}$. This observation approximately agrees with the estimation derived from the theoretical model ($Q = 4.8\,\mu W\mu m^{-2}$).

We characterized the distortion of the interferometric image of the scattered wave associated with the thermo-induced phase-shift of the reference wave. We used a 2D Gaussian fit to estimate the positions of the interferometric minima (corresponding to a heating power of 40 mW) and the interferometric maxima (corresponding to a heating power of 110 mW). By comparing the absolute change in the estimated positions of the images of three different nanospheres, we found that shifting the phase of the reference beam by π distorts the image by <10 nm in all directions (Fig. 2e). We attribute the marginal image distortion to two effects (i) a possible asymmetry of the point-spread-function having a different effect on the peak and dip localization and (ii) possible thermal distortion of the glass substrate owing to an inhomogeneous heating of its surface. Our estimation indicates a possible small bent of ~30 arcsec owing to the thermal expansion. Nevertheless, these effects combined result in distortion of <0.5% of the field of view, allowing to adjust the contrast of a scattering object (e.g., a label) and follow its position in three dimensions. The measured distortion does not compromise the localization precision and the minimum measurable increment in the 3D position remains unaffected (Supplementary Fig. 7).

The temporal change of the image contrast in response to a 100 Hz square modulation is illustrated in Fig. 2g. The phase change requires up to 1 ms to reaches 96% of the steady-state level (rising time of 0.94 ± 0.2 ms and falling time of 1.0 ± 0.2 ms), which is consistent with the theoretical model in Fig. 1i. The maximum rate of the phase change is limited by the heat propagation through the glycerol layer yielding $10 \pm 2$ mrad μs$^{-1}$. The temporal evolution of the contrast is not a single-exponential function due to different rates of heat dissipation in the materials of the phase modulator structure. In particular, upon heating up, a thermal gradient is built in the liquid thermo-optic material and at a slower rate generates a thermal gradient in the glass substrate. Conversely, the cooling of the thermo-optic material is approximately equally fast as the heating, however, extracting the heat accumulated in the glass substrate result in slight inertia of the phase shift with a slower gradient towards the equilibrium. Therefore, even at 1 kHz phase modulation, we were able to achieve 77% of the maximum phase modulation (Fig. 2h).

To exploit the maximum rate of the phase change offered by the geometry of the PT-SLM, we optimized the temporal profile of the heating intensity, as shown in Fig. 2i. For the heating, the temporal heat profile begins with a 100-μs pulse with a heating laser power of 110 mW, followed by a sharp decrease in the heating laser power (83 mW) to stabilize the temperature increase towards a steady state. For the cooling, a 100-μs pulse (0 V) was followed by an increase of the heating level (69 mW) for stabilization. Based on the analysis of 100 modulation cycles, we achieved the mean rise time of $0.2\pi$ phase switching as short as $70 \pm 6$ μs (defined here as 20–80% of the magnitude of the signal).

The stability of the phase modulation is characterized in Fig. 2j yielding a standard deviation of 5 mrad with no heating applied, which corresponds to the shot-noise-limited baseline, and 8 mrad with the heating on showing a combined effect of the shot noise and the heating laser stability. We characterized the reproducibility of the PT-SLM at 500 Hz modulation and extracted equilibrium levels of contrast in each modulation cycle (Fig. 2k), resulting in a standard deviation of 5 mrad for both 0 and $\pi$ phase-shifts. We incorporated a $\lambda/2$ waveplate before the PT-SLM to characterize the contrast change dependence on polarization between 0 and 180° in Fig. 2l. As the heat-induced phase shift is intrinsically isotropic, we attribute the residual fluctuation of <3% (30 mrad, standard deviation) merely to the inhomogeneity of the polarization optics. A comparison of the stability, reproducibility, and polarization uncertainly in a box-plot representation is shown in Fig. 2m.

**Quantitative phase imaging of nanoscatterers.** Having direct control of the interference contrast with the PT-SLM, we developed a method of quantitative phase imaging of light scattered by nanoscopic scatterers. We collected a series of images of a single scatterer while modulating the phase-shift of the reference wave, and extracted the terms $s/r$ and $\Delta\varphi$ from contrast variation using least-square fitting of Eq. (3) (details in Supplementary Note 4). The images of two gold nanospheres (30 nm) at different focus positions and calculated squared amplitude $(s/r)^2$ and a phase image are shown in Fig. 3a–c. A plot of the variation of the phase at the centers of the two images of nanoparticles as a function of the focus displacement over 1.5 μm is shown Fig. 3d. The phase profile corresponds to the Gouy phase of a focused coherent optical beam, which is known to follow an arctangent function for a Gaussian beam; however, in the image of an optical microscope with a circular aperture, the theory suggests a linear dependence on the focus position[42]. The theoretical Gouy phase profile closely matches the slope of the experimental data as shown in Fig. 3d,

following the dependence

$$\mathrm{d}\varphi_{Gouy}/\mathrm{d}z = ka^2/4f^2,  \quad (4)$$

in which $k = 2\pi/\lambda$ is the wavenumber, $a$ is the radius of the aperture, and $f$ the focal length of the microscope objective. The residual deviation of the experimental curve from the linear theoretical model indicated more specific field distribution at the microscope objective aperture.

The phase stability and reproducibility of the PT-SLM allow for highly sensitive and reproducible extraction of the phase information. To demonstrate the phase extraction sensitivity, we imaged a single 20 nm single gold nanoparticle immobilized on the coverslip surface with a series of phase-sweeps (580 cycles at 500 Hz modulation) and extracted the scattering phase of the nanoparticle at the center of its diffraction-limited spot for each heating and cooling phase-sweep. We explored the effect of the position-dependent PSF distortion on the precision of the phase extraction in Fig. 3e by moving the same nanoparticle to five different positions within the field of view (indicated in the inset) using a $xy$ piezo stage. The box plot in Fig. 3f depicts the cycle-to-cycle variation in the measurement of the scattering phase at the five different nanoparticle positions. We observed an average noise-limited standard deviation of 13 mrad, which is translated to an uncertainty of 0.4 nm of the vertical position. Therefore, the vertical displacement resolution is close to the localization precision estimated from the reconstructed amplitude images as characterized in Supplementary Fig. 7.

**High-sensitivity QPI of single microtubules.** To demonstrate the usability of the nanoscopic QPI method for biological nanostructures, we imaged two microtubules, shown in Fig. 4a, overlapping on an atomically flat mica surface supported by a coverslip[14]. The specific geometry of the sample allowed us to separate the images of the two microtubules. We averaged the cross-section pattern of each of the microtubules and subtracted these patterns from the original image to obtain the contribution of a single-microtubule as shown in Fig. 4b, c (details in Supplementary Note 5). The local fluctuation of the microtubule scattering phase indicates changes in the distance $z$ between the microtubule and the surface. To reconstruct the 3D profile of microtubule crossing in ribbon-like surfaces (Fig. 3h), we extracted the vertical coordinate of the microtubule scattering as $z = \frac{\lambda}{4\pi n_m}\triangle\varphi$, in which $\lambda = 488$ nm is the wavelength, $\triangle\varphi$ is the change in the detected phase and $n_m = 1.33$ is the refractive index of the surrounding medium. In the area of microtubule crossing, the light scattered on the microtubule segment further away from the surface passes through the underlying microtubule. Considering the microtubule diameter of 25 nm and refractive index of the microtubule of approximately 1.48,[9] we estimated the effective refractive index within the diffraction-limited area of the microtubule crossing to be $n_{eff} = 1.35$. Therefore, the contribution of the underlying microtubule to the measured phase-shift of the light scattered from the upper microtubule is <2%, which is smaller than the measurement uncertainty and, thus, negligible. The height profiles plotted along the axes of the two microtubules indicate that one of the microtubules, the green ribbon corresponding to Fig. 4d, features a vertical displacement at the location of the microtubule crossing. The displacement peaked at 40 nm, which corresponds to the maximum elevation of the microtubule in the area where the microtubules overlap. In the direction along the microtubule axis, the profile forms a broad peak with a full width of 500 nm, which is larger than the microscope point-spread function. We carried out a control experiment using an atomic force microscope (AFM) of an equivalent microtubule structure shown in Fig. 4e. The control

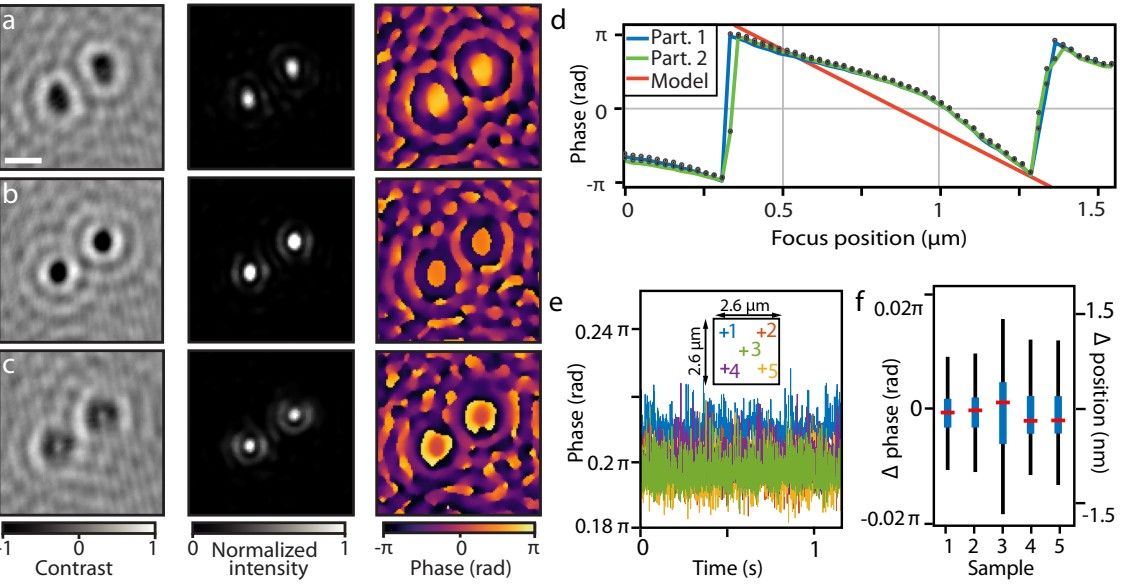

**Fig. 3 Quantitative phase imaging of gold nanoparticles. a–c** From left to right: iSCAT contrast, reconstructed intensity, and phase image of two gold nanoparticles at the focus positions: (**a**) 0.35 μm (coverslip closer to the microscope objective), (**b**) 0.6 μm (~best-focus position) and (**c**) 0.85 μm (coverslip further away from the microscope objective). Representative images out of 50 individual nanoparticles imaged, scale bar indicates 500 nm. **d** Focus position dependence of the phase for the two particles shown in (**a–c**). A theoretical prediction is shown in red. **e** Time series of the extracted scattering phase of a 20 nm gold nanoparticle at five different positions (indicated in the inset). **f** Box plot of the variation of the reconstructed phase at the five positions of nanoparticles in (**e**) (offset to the mean value of all five positions) and the corresponding vertical displacement (Δposition). The interquartile range is in blue, max–min range in black, and the median in red ($n = 60$ phase reconstructions for each position)[53].

experiment shows a strikingly similar geometry of the longitudinal transition region of the microtubule crossing (36 nm displacement peak of 500 nm width). A more detailed analysis of the shape corrugation of a microtubule attached to the surface using different densities of kinesin linkers is described in Supplementary Note 6.

**Fast-tracking of single scattering labels with full phase reconstruction.** The short response time of the PT-SLM approach offers a number of unique applications for analyzing the dynamics of biological systems in great detail. The direct and fast access to the scattering phase allowed us to perform a 3D localization of weak scatterers used as molecular labels and describe their interactions and motion at very high rates without the need for any additional calibration step for the quantification. We studied the interaction of a microtubule crosslinker protein Ase1/PRC1 with the microtubule. We immobilized double-stabilized microtubules on the microscope coverslip, decorated them with His-tagged Ase1 molecules, and anchored 20-nm gold nanoparticles specifically to the His-tag in the dimerization region of Ase1 molecules (Methods). We imaged the gold nanoparticle at 200,000 frames per second and every 1 ms, we swept the phase of the reference beam using the PT-SLM to extract the quantitative phase information of the scattered light. We used the reconstructed intensity image to localize the position of the nanoparticle label. We used the reconstructed phase image to quantify the immediate distance of the nanoparticle from the glass surface. As a result, we obtained the 3D trajectory of the nanoparticle at a 1 kHz rate, shown in three different projections in Fig. 5a–c and reconstructed in the Supplementary Video 1. Interestingly, the *xy* localization data rendered a ribbon-like area of ~80-nm width (Fig. 5a). The projection perpendicular to the microtubule in

Fig. 5c indicates that the highest density of 3D localizations (color-coded in the scatter plot) forms an arc, which we interpreted as a segment of a cylindrical surface with a radius of ~40 nm. The size of this experimentally observed radius can be explained as the combination of the 12.5 nm radius of the microtubule with the ~20-nm long Ase1 crosslinker[43] and the 10 nm radius of the gold nanoparticle. The uncertainty in the radial coordinate of the cylindrical trajectory of $\sigma_\rho = 7$ nm is larger than the localization precision in any direction or the phase sensitivity. Considering the long Ase1 crosslinker and highly diffusive character of the scattering label, we attributed the radial uncertainly to the true fluctuation in the label position. The cylindrical trajectory was truncated at ~25 nm above the coverslip surface (Fig. 5c), which we attribute mostly to the thickness(~7 nm) of the bovine serum albumin (BSA) cushion surrounding the microtubule[44] combined with the nanoparticle radius of 10 nm, the Debye length of <1 nm and the fluctuation in the particle position within the 1-ms time window.

The observation of the 2D diffusion of the microtubule crosslinker on the surface of the microtubule is a striking piece of evidence that would not be possible to measure without achieving the kHz data acquisition speed. In stark contrast, if the very same trajectory was measured at a conventional video-rate of 30 frames per second[9,45], as simulated by the down-sampled and smoothed example in Fig. 5d, the conclusion would be that the Ase1 molecule is within an 8-nm distance from a single microtubule protofilament suggesting false, although often anticipated behavior.

The fact that the Ase1 molecule diffuses randomly in the longitudinal as well as perpendicular direction governs the delocalization of the molecule around the whole perimeter of the microtubule. This character of the molecular diffusion increases the probability of finding a region of a point contact between two

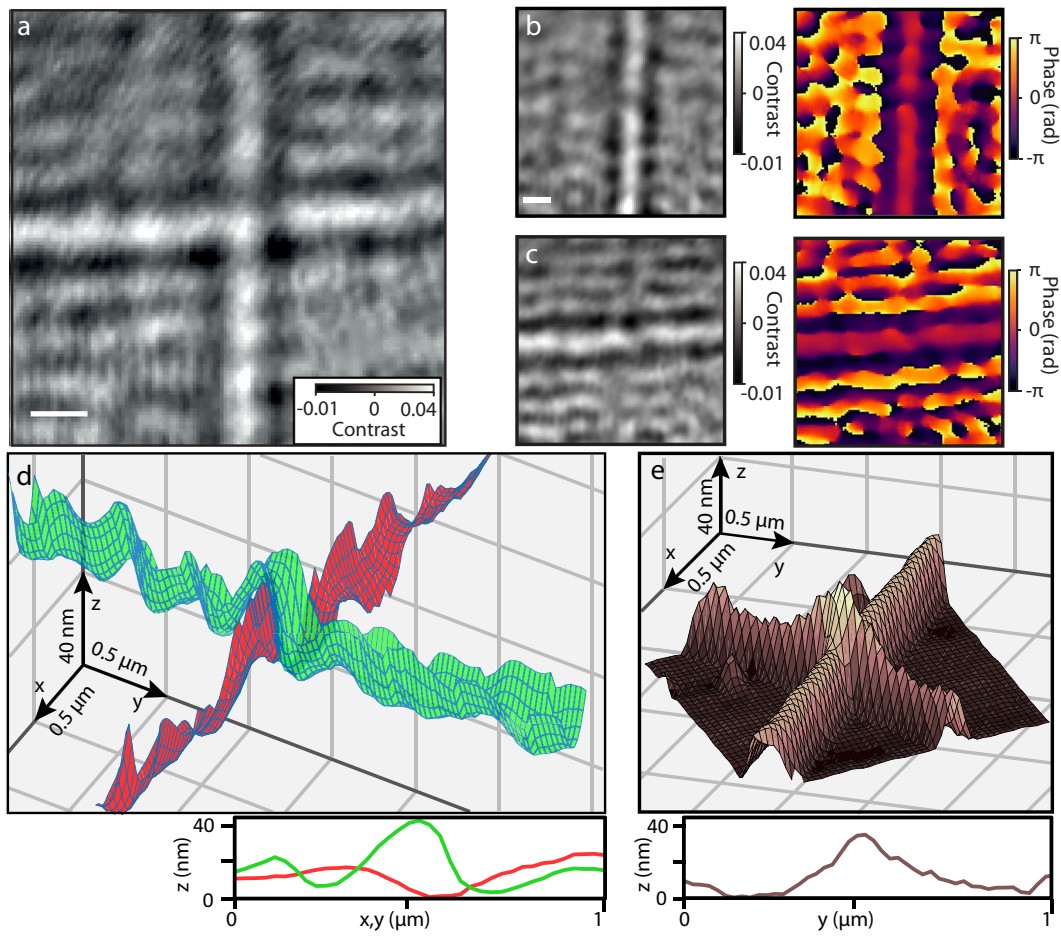

**Fig. 4 Quantitative phase imaging of microtubules. a** iSCAT image of two microtubules. Representative image from five independent experiments, scale bar indicates 500 nm. **b–c** Separated iSCAT images and phase images of individual microtubules shown in (**a**). **d** 3D reconstruction of the crossing microtubules and the cross-section height profile. **e** AFM image of microtubule crossing and the corresponding height profile[53].

microtubules, detailing the mechanism of the diffusive microtubule crosslinkers navigation through complex microtubule networks. Control data obtained using single-molecule fluorescent microscopy, provided in Supplementary Fig. 1, show that a microtubule crosslinker protein can transit from one microtubule to another in the region of the microtubule crossing.

## Discussion

We demonstrated a fast and highly spatially confined phase adjustment of a free-space optical beam using a PT-SLM module based on the photothermally driven thermo-optic effect. The advancement of the PT-SLM resulted from the optimized thin layer of the thermo-optic material and increased rate of heat dissipation, which improved the spatial resolution and modulation speed and suppressed the effect of the thermal lens at the edges of the heated area. We combined the PT-SLM with an interferometric scattering microscope to specifically address the reference arm of the common-path interferometer and performed quantitative phase imaging with exceptional nanoscopic sensitivity. It is worth mentioning that iSCAT imaging is highly sensitive to any wavefront perturbation. To date, any wavefront shaping involving LC-SLM has failed to be integrated with iSCAT owing to insufficient phase stability and diffraction speckles. The PT-SLM presented here has proved to be compatible with the most demanding iSCAT applications as modulations occur exclusively at the lowest spatial frequencies of the scattering beam (0.25% of the beam area). A laboratory prototype of the phase

modulator features a phase stability in the milliradian range (~ 0.2% of the modulation amplitude) with marginal distortion of the optical image of <0.5% (10 nm), it works with high transmittance (>80%) and suffers from no grating or polarization effect. In our proof of concept experiment, we achieved a switching time of the phase as fast as 70 μs, which is two orders of magnitude faster than conventional LC-SLM devices and one order of magnitude faster than previously reported thermo-optic spatial-light modulation techniques[37]. We argue that thanks to this unique concept of the photothermal phase modulator we could efficiently extract the scattering phase information. Indeed, we demonstrated the sensitivity of the nanoscopic QPI to accurately characterize the scattering phase of single gold nanoparticles with a precision of 10 mrad corresponding to the vertical resolution of 0.4 nm. These improved characteristics enabled the 3D mapping of single microtubules crossing and the 3D localization of a single protein labeled with a small scattering label. The quantitative 3D trajectories of microtubule cross-linkers acquired at the kHz rate revealed a complex trajectory of the protein diffusion on the surface of microtubules explaining the phenomenon of the protein navigation through microtubule networks. The achieved level of performance offers faster and more accurate wave-shaping and pushes the limits of the QPI deep into the subdiffractional regime. Indeed, we foresee numerous other applications that can exploit the potential of this PT-SLM technique, ranging from microscopy, biomedical imaging, or digital holography, to astronomy.

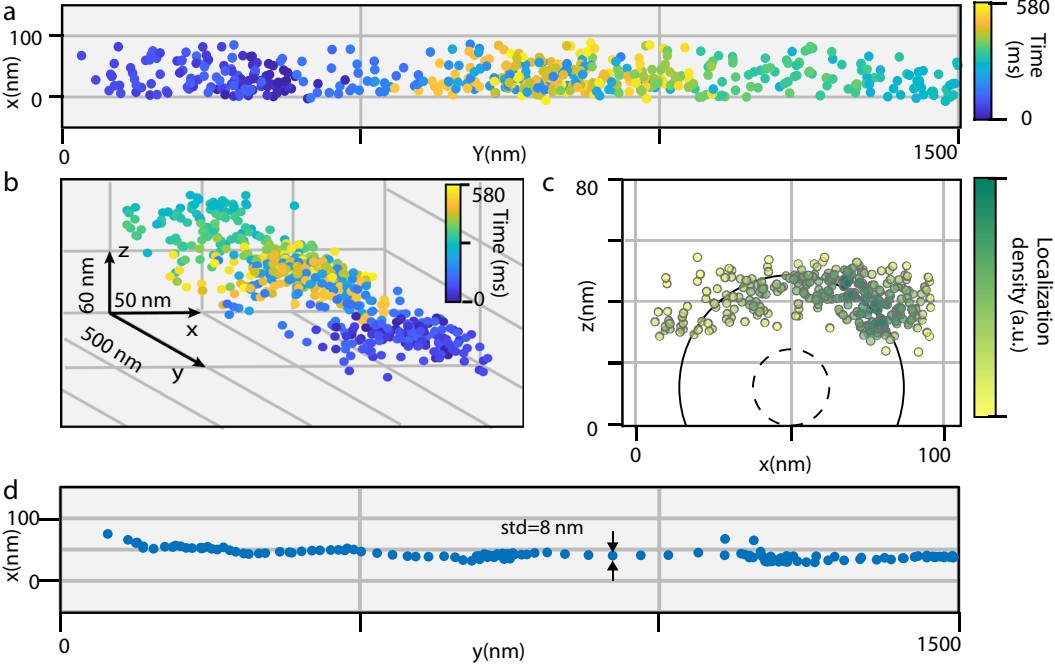

**Fig. 5 3D localization of an Ase1 protein on a microtubule using quantitative phase imaging. a** Scatter plot of the protein trajectory in the *xy* projection, and (**b**) 3D perspective view. **c** The cross-section scatter plot of the trajectory in the *xy* plane (perpendicular to the microtubule); density of localizations is color-coded. The approximate position of the microtubule (radius 12.5 nm) is indicated with the dashed-line inner circle and the estimated trajectory radius of 40 nm is indicated with a solid line. **d** The protein trajectory after smoothing and down-sampling to the rate of 30 samples per second[53].

## Methods

**Simulation parameters**. For the simulations presented in Fig. 1, we considered a disk heat source with a diameter of 60 μm, delivering heating power ranging from 0.4 to 14 μW μm$^{-1}$, a glycerol layer thickness $L = 5$, 20, and 250 μm with a thermal conductivity $\kappa = 0.285$ Wm$^{-1}$K$^{-1}$, a 140 μm thick glass BK7 layer with $\kappa = 1.2$ Wm$^{-1}$K$^{-1}$, and a 1 mm sapphire layer with $\kappa = 27.21$ Wm$^{-1}$K$^{-1}$. To calculate the refractive index variation of glycerol, we used the first order of the Taylor expansion coefficient: for glycerol $\triangle n / \triangle T = -2.7 \times 10^{-4}$ K$^{-1}$, for BK7 glass $\triangle n / \triangle T = 3 \times 10^{-6}$ K$^{-1}$ and sapphire $\triangle n / \triangle T = 13 \times 10^{-6}$ K$^{-1}$. Simulations were processed with the Matlab software.

**Synthesis of gold nanorods**. Materials used: tetrachloroauric acid (HAuCl4 ,3H2O), sodium borohydride, silver nitrate HCl, NH4OH (32%), Poly-(styrenesulfonate) (PSS, Mw 70,000, 20 wt%) and Poly(diallyldimethylammonium chloride) (PDDA, MW < 100,000), were purchased from Aldrich. Poly(vinylpyrrolidone) (PVP, MW 40,000) was supplied by Fluka. Ascorbic acid, sodium chloride (NaCl), cetyltrimethyl ammonium bromide (CTAB), poly(diallyldimethylammonium chloride) (PDDA, MW 100,000, 20 wt%) were procured from Sigma. All chemicals were used as received. A total of 96% pure isopropanol and Milli-Q grade water were used to make up all solutions.

The gold nanorods synthesis is adapted from the protocols developed by Nikoobakht et al.[46] Liu et al.[47,48]. All the glassware and the magnetic stirrers were first cleaned with an aqua regia solution (HCL/HNO3 – 70/30) for 5 min, then washed with Milli-Q water and isopropanol and dried. The gold seed were synthesized by borohydride reduction of 0.28 mM HAuCL4 in an aqueous 0.1 M CTAB solution. 76 μL of the seed solution was added to following growth solution (8 mL): 0.1 M CTAB, 0.5 mM HAuCL4, 0.8 mM ascorbic acid, and silver nitrate (0.08 mM). The solution was then put at 30 °C overnight. The excess CTAB solution was removed by centrifuging (8000 rpm, 20 min) and redispersing in Milli-Q water. Then the nanorods were functionalized with PVP by carefully pouring drop by drop the nanorods solution in a PVP solution under vigorous stirring. After a night of gentle stirring, the excess PVP solution was removed from the nanorods solution centrifuging at 4000, 5000, and 7000 rpm for 10 min and each time redispersed in isopropanol under sonication.

One hundred and seventy μm thick cleaned coverslip glass were coated with the gold nanorods[49]. Coverslips were immersed for 20 min in a 1 g/L PDDA and 0.5 M NaCl solution. After rinsing with Milli-Q water and a drying, the coverslips were dipped for 10 min in 1 g/L PSS and 0.5 M NaCl solution followed by 10 min in a 1 g/L PDDA and 0.5 M NaCl solution. The functionalized coverslips were immersed in a solution of PVP gold nanorods for 3 h. The immobilization was promoted by the electrostatic interactions between the positively charged coverslip and the negatively charged gold nanorods.

**Photothermal SLM module preparation**. To confine the liquid glycerol thickness, we placed a droplet of 10 μL of glycerol on the layer of gold nanorods and sandwiched the droplet with a sapphire window. The thickness of the glycerol layer was estimated from the area wetted by the glycerol layer, yielding ~20 μm.

**Microscope sample preparation**. Coverslips were washed in ethanol and cleaned using a plasma cleaner (Zepto-BLS, Electronic Diener). The 30 nm gold nanospheres (BBI Solutions) were directly spin-coated on a glass coverslip. For the crossed microtubule experiments (iSCAT and AFM), we immobilized microtubules stabilized with GMPCPP (guanosine-5′-[(α,β)-methyleno]triphosphate, a non-hydrolyzable analog of guanosindiphosphate) and Taxol[50] to a positively charged surface of mica glued to the glass coverslip[14]. We flowed the microtubule solution through an X-shaped flow chamber consecutively at the two perpendicular flow directions.

**Optical setup specification**. The imaging light source used in the iSCAT experiment was a continuous laser iBeam smart, TOPTICA Photonics AG, $\lambda = 488$ nm. A microscope objective αPlan-Apochroma Zeiss x100, $NA = 1.46$ was used to image the sample on a CMOS-based camera from Photon focus, MV1-1024E-160-CL at a total magnification of x333. For fast tracking experiment iX Camera i-Speed 508 was used. The heating light source was a continuous laser Cobolt Flamenco, $\lambda = 660$ nm modulated with an acousto-optic modulator from AA Opto-Electronic, MT90-A1 VIS. A piezo stage was used to manipulate the sample.

**AFM characterization of microtubules**. The morphology of the microtubules immobilized on the glass surface was measured using an atomic force microscope NanoWizard3 (JPK instruments AG). A special cuvette with a glass coverslip bottom functionalized with poly-L-ornithine was filled with BRB80 buffer followed by injection of GMPCPP microtubules using a micropipette. A force spectroscopy mode (QI$^{TM}$ mode) of the AFM and qp-Bio AC cantilevers (Nanosensors, Switzerland) were used to probe the crossed microtubules with force constant setpoint down to 0.1 N/m to prevent mechanical damage of the microtubule.

**Flow chamber preparation**. 18 × 18 mm (top) and 22 × 22 mm (bottom) coverslips were rinsed with ethanol, water, dried with N₂, and cleaned with oxygen plasma (Diener, Germany). First, the bottom coverslips were dipped in acetone for 10 s and then immersed in the APTES solution (2% wt. solution of 3-aminopropyl triethoxysilane in acetone) for another 10 s. Finally, the bottom coverslip was rinsed with acetone and dried with N₂. The top and bottom coverslips were assembled with two parallel spacers of Parafilm M and heated for sealing the chamber.

**Ni-ANTA functionalization of gold nanoparticles**. 20-nm gold nanoparticles (GNPs, BBI solutions) were cleaned four times by centrifugation at 8000 rpm for 10 min and resuspended in 20 mM 4-(2-hydroxyethyl)-1-piper-azineethanesulfonic acid (HEPES) buffer with 0.1% polyethylene glycol sorbitan monolaurate (Tween-20). These pre-cleaned GNPs were incubated with 0.65 mM carboxythiol solution (HS-C11-EG6-OCH2-COOH, Prochimia) and left overnight. The unbound thiol molecules were removed by 3 cycles of centrifugation at 6000 rpm for 10 min and resuspended with 20 mM HEPES buffer with 0.1% Tween-20. Next, the carboxyl groups were activated with 0.45 mM N-hydroxysuccinimide (NHS) and 1.70 mM N-(3-Dimethylaminopropyl)-N′-ethylcarbodiimide hydrochloride (EDC) for 5 min[51]. Activated GNPs were incubated with 0.40 µM of Ni-ANTA. Ni-ANTA was prepared before by stirring 100 µL of 100 mM of Nα,Nα-bis(carboxymethyl)-L-lysine hydrate (ANTA) solution with 100 µL of the 50 mM NiCl2 solution for 90 min. After 2 h, the unreacted carboxylic groups were deactivated with 125 mM of ethanolamine. Finally, the GNPs were washed twice at 5000 rpm and resuspended in 20 mM HEPES with 0.1% Tween-20.

**Ase1 experiment**. The flow chamber was mounted on the microscope and filled with BRB80/Tx (80 mM piperazine-N,N′-bis(2-ethanesulfonic acid) (PIPES), 1 mM MgCl$_2$, 1 mM ethylene glycol-bis(2-aminoethylether)-N,N,N′,N′-tetraacetic acid (EGTA), 10 µM Taxol, pH 6.9). We injected 10 µL of double-stabilized microtubules and checked their concentration on the surface with the microscope. The rest of APTES surface was passivated for 5 min with the BSA solution (10 mg/ml BSA in BRB80/Tx). The BSA solution was flushed with the motility buffer (20 mM of HEPES, 2 mM of MgCl$_2$, 1 mM EGTA, 37.5 mM KCl, 10 µM Taxol, 0.5 mg/mL casein, 10 mM DL-dithiothreitol (DTT), 1 mM adenosine 5′-triphosphate (ATP) (+Mg), 20 mM D-glucose, 220 µg/mL glucose oxidase and 20 µg/mL catalase, pH 6.9). Then, 20 µL of Ase1-GFP (green fluorescent protein) protein were injected (0.9 nM in the motility buffer). After 2 min, 10 µL of Ni-ANTA GNPs were added to label Ase1-GFP protein. Before the measurement, the flow chamber was flushed with the motility buffer to reduce the background fluctuations.

**PRC1 fluorescence crossover experiment**. Flow chambers were assembled using two cleaned and silanized (0.05% dichlorodimethylsilane in trichloroethylene) glass coverslips. The chambers were first washed with PBS (phosphate buffered saline), followed by incubation with 10 µg mL$^{-1}$ anti-tubulin antibody (Sigma Aldrich, T7816) in PBS for 10 min prior to surface passivation by 1% Pluronic F-127 in PBS for at least 1 h. Next, the chambers were washed with BRB80/Tx, microtubules were introduced and unbound microtubules were removed after 1 min with BRB80/Tx. Next, the solution in the chamber was exchanged by motility buffer (BRB80 containing 10 µM Taxol, 10 mM DTT, 20 mM D-glucose, 0.1% Tween-20, 0.5 mg/mL casein, 1 mM ATP(+Mg), 0.22 mg/mL glucose oxidase and 20 µg/mL catalase), followed by the introduction of 0.25 nM full length mRuby-labeled PRC1 (the human orthologue of the *s. pombe* Ase1) for single molecule imaging mRuby-PRC1 was imaged using the total internal reflection fluorescence (TIRF) mode of an inverted widefield fluorescence microscope (Nikon Eclipse Ti2; Nikon, Tokyo, Japan) equipped with a motorized *xy* stage, a perfect focus system, a 60x oil immersion objective (Nikon Apo TIRF 60x Oil, NA 1.49, WD 0.12 mm), a Prime BSI Scientific CMOS camera (Teledyne Photometrics, Tucson, AZ, USA) and an EM500-545 filter turret. Fluorescence images were acquired for 60 s at 200 ms exposure time using NIS-Elements Advanced Research software v5.02 (Laboratory Imaging, Prague, Czech Republic).

**Image acquisition and data processing**. To discern the scattering of the investigated specimen, all images were normalized to the illumination pattern for each focus position and each heating power. The reference illumination pattern was averaged from raw images acquired while the sample was moved in a circular motion with a diameter of 5 µm. To characterize the residual image distortion the apparent image positions of nanoparticles were localized by 2D Gaussian fitting using Fiji Java 8 version 1.8.0_66 from ImageJ and the plugin TrackMate[52]. Amplitude and phase images were calculated using the Matlab2018a software.

**Biological materials availability**. PRC1, Ase1, and tubulin are readily available from the authors and other biological materials are available from Sigma Aldrich.

**Reporting summary**. Further information on experimental design is available in the Nature Research Reporting Summary linked to this paper.

## Data availability
All data supporting the findings of this study are available on the Figshare repository https://doi.org/10.6084/m9.figshare.14170622.

## Code availability
The custom code for the data acquisition software is device-specific and available from the authors on reasonable request. The code used to perform the theoretical simulations and the code for the calculation of the phase image is available on the GitHub repository under GNU General Public License, version 3.0 (GPL-3.0): https://github.com/HRNanoOptics/Fast-photothermal-spatial-light-modulation-for-quantitative-phase-imaging-at-the-nanoscale-code.

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

## Acknowledgements

This work was funded by the Ministry of Education, Youth and Sports of the Czech Republic under the project LL1602 and by the Czech Science Foundation under the project 18-19705 S. The authors thank Guillaume Baffou, Nicholas Scott Lynn, and Xavier Audier for scientific discussions.

## Author contributions

H.M.L.R. and M.P. conceived the research, H.M.L.R. derived the theory, carried out the experiments, processed and interpreted the data, K.H. developed the microtubule assay and the Ase1 experiments, L.B. developed the experiment control and acquisition software, M.V. carried out the AFM study, V.H. purified proteins and performed fluorescence microscopy experiments, H.M.L.R., Z.L., and M.P. wrote the paper.

## Competing interests

The authors declare the following competing interests: H.M.L.R. and M.P. of the Institute of Photonics and Electronics CAS have a patent related to the method of phase modulation. The remaining authors declare no competing interests.
