## [Peer Review File · Nature Communications]

Reviewers' comments:

Reviewer #1 (Remarks to the Author):

This paper describes the use of a novel photothermal-based phase-modulation That exploits laser heating of an optical element constructed from the layer-by-layer deposition of gold nanorods. This device is used to impart phase control for quantitative phase imaging; demonstrated here via interferometric scattering microscopy of microtubules.

Although photothermal modulation of refractive index has previously been reported (e.g. Heber ACS Nano 2014, 8, 2, 1893–1898), this paper provides a route to spatial control of the phase profile. The authors also report an impressive response time in this proof-of-concept apparatus of 70 us, and estimate a theoretical minimum of 430 ns.

The potential advantages of this method are clear, there remain a number of technical hurdles that have not been overcome in these current experiments, and do limit its utility: (1) The spatial light modulation demonstrated by the authors is of a fixed circular phase mask, and is not used as an active optical element. (2) The authors demonstrate a 70 us switching time upon heating of the phase element. However, to be really useful, it is fast switching on and off that is required for high speed phase imaging. Cooling takes longer, of the order of 2 ms in the current device. (3) The example application demonstrated by the authors is microtubule iSCAT imaging. It is clear that this is intended as a demonstration of their phase element's capabilities, but worth noting that no specific advance in our understanding of microtubule biology was expected. Thus the impact of the work must be judged on the novelty and importance of this new thermally-driven phase element.

For spatially homogenous phase elements, even simple piezo-driven mirrors have resonant frequencies of many 10's of kHz, faster than the overall framerate that this current method could achieve. If the improved microsecond step response time had, for example, been used to measure a physical or biological phenomena its impact would be significantly higher.

There are a number of other more minor considerations that I hope the authors might address. Most pressing is their description of separating the interferometric signals from the two crossing microtubules shown in figures 4 and S7. We are left to guess precisely how these two signals were 'separated', from S7 it looks like x and y components of the original image were analysed separately - relying on the 90 degree crossing in this particular image, but this needs to be made clear.

Reviewer #2 (Remarks to the Author):

Robert et al. suggest and demonstrate a polarization-insensitive and fast switching SLM, based on the photothermal effect. The approach, in my opinion, could have strong potential in microscopy due to these inherent advantages, and the paper is well written. Disappointingly, the biggest potential advantages of this SLM – insensitivity to polarization, and fast switching - are not demonstrated experimentally. In this sense, this paper presents a limited proof of concept; all the presented experiments could have been performed using existing well-established methods, e.g. a deformable mirror. This is without mentioning the limited spatial resolution that is demonstrated (a single disk of 60 um diameter).

Specific comments/suggestions:

1. The results in this manuscript would be much stronger if the authors could demonstrate an application that gains from the unique potential of their beautiful photothermal SLM concept. Probably the easiest demonstration to perform would use the fast switching aspect.

2. A clearer statement of novelty would help. What exactly are the differences between past photothermal SLMs and the current one?
3. The microtubule experiment raises a few questions:
 - a. What is the value of "n" used to calculate the z value from the phase? Is n of the microtubule assumed to be equal to n of the environment? In the crossing, for example, there is a microtubule under another one. Is this accounted for? Or assumed negligible?
 - b. The authors report: "vertical displacement at the location of the microtubule crossing peaking at 40 nm with a full width of 500 nm". This number, 500 nm is even validated using AFM on a similar (?) crossing. But what exactly is 500 nm wide? The microtubule (unlikely)? It is not clear.
4. I am confused by the following: the text states: "The temporal heat profile begins with a 100- μ s pulse with a heating laser power of 110 mW. It is followed by a sharp decrease in the heating laser power (44 mW) to stabilize the temperature increase towards a steady state." But from fig. 2h it seems the HLP goes to 0.
5. Figure 3d: which phase is plotted? In the center of the nanoparticle?
6. Figure S8a – how long is the scalebar?
7. A major application that uses SLMs and would benefit from the proposed concept is fluorescence-based 3D single molecule localization microscopy and particle tracking. This could be included in the introduction as motivation along with relevant references.

Reviewer #3 (Remarks to the Author):

Robert et al give a proof-of-concept demonstration of a novel photothermal spatial light modulator to introduce a phase shift between scattered and reference beams in a common-path iSCAT microscope. This permits them to perform quantitative phase imaging (QPI) on matter such as gold nanocolloids and microtubules, and on the latter the phase information is used for 3D reconstruction of the scatterer.

The paper overall is well-written and structured in a clear manner. To their credit, the authors provide plenty of quantitative experimental details and thorough characterization of the photothermal spatial light modulator (PT-SLM). The authors also provide a wealth of supplementary information which helps support the claims in the main manuscript, and it is clear a great deal of care and attention has been paid in the creation of this work as well as the production of the manuscript. The paper is a pleasure to read.

The combination of fast QPI with iSCAT is an interesting idea. While neither QPI or interference microscopy of nanomaterials is new, their successful combination provides fertile ground in what I believe would be of interest and practical use within communities such as nano-optics and microscopy, as well as single molecule/particle (bio)imaging.

My main concern, however, is that as the manuscript stands, I do not feel the authors have demonstrated a sufficient level of progress that would convincingly enable themselves new lines of scientific inquiry (e.g. nanometric QPI of uncharacterized or dynamic samples) or motivate interest and influence other in the field to adopt the method presented. This is not to take away from the accomplishments of the authors, indeed the work presented is quite impressive and the thorough level of care taken leads to one to be convinced of the exciting possibilities of QPI-iSCAT. My reservation with endorsing the manuscript is that I do not see demonstrated enough advance in progress over what else exists in the literature that is warranted in such a high-impact journal or expected by the community to open up real lines of new experimental inquiry. I will outline my specific main concerns below and follow with some minor questions.

The device demonstrated (PT-SLM) is smart and inventive solution to introducing a phase-shift for a common-path interferometer. The extensive modeling and characterization of the device by the authors is a strong point, and the performance claimed by the authors is convincing. However, the

device by itself does not merit publication within such a journal since it does not consist of any new discoveries, only a synthesis of existing materials and technologies, albeit in a novel and clever way to tackle an interesting and useful problem.

Since the authors introduce the device for the application of QPI “at the nanoscale”, this claim requires scrutiny.

The authors claim the PT-SLM can operate quickly and also permit QPI with exceptional nanoscopic sensitivity. These two important claims need to be more strongly and convincingly evidenced in practice for the work to be justifiably considered to advance the progress of the field.

Firstly, the speed by which the full phase can be swept, and hence sampled, is exciting, and hence would be something the community would want to see – and use! I would like to see a demonstration of a time-dynamic nanoscale sample being imaged fast with full phase information. For example, Kandel et al (DOI: 10.1021/acsnano.6b06945) perform SLM-QPI microscopy to image microtubule mobility at video frame rates. I do not see the authors demonstrating an improvement in imaging performance (speed and sensitivity) beyond what was achieved in that work.

Secondly, besides how fast one can extract full-phase information, I believe what is also important to this work is the sensitivity one can perform QPI on nanoscale scatterers. This essentially reduces to how well one can determine the phase from such a weak scatterer. This is where I believe iSCAT enters, and I would want to see a more convincing demonstration of the superior sensitivity that iSCAT is capable of (for example, as demonstrated in previous works of the corresponding author) being applied to give “exceptional” nanoscopic sensitivity in QPI and 3D morphology reconstruction. The measurement surrounding Figure 4 is good, and the authors to their credit include AFM measurements in corroboration of their QPI of the microtubule. I accept the authors claim to resolve the crossing of two microtubules, with reconstruction of their height plausibly showing one lying upon another. My concern however is that the reconstructed surface map appears quite noisy, and so I would always have trouble knowing whether the other nanoscale bumps on the tubule are real or noise – hence this is not so practically useful. I appreciate the authors argue the morphology is related to the surface preparation (Supporting Information 8), but no direct evidence is given that their imaging is sufficiently precise for these features to be real and not simply noise.

This lack of sensitivity is, I feel, important as the QPI image (Fig.4d) is not compelling by itself, nor is it reliable for nanoscale characterization, and only can be confidently interpreted when supported by a separate measurement such as the AFM image (Fig.4e). For that reason, I do not feel the authors have demonstrated sufficient progress to convince that truly reliable and sensitive nanoscale QPI has been achieved in this work, which I would say is required for publication here. I would want the authors to discuss the issue of detection sensitivity with their normal excellent level of care, detail and attention. They do not discuss the uncertainties of their phase reconstruction, nor discuss how good their imaging could be in regard to nm-sensitivity in the reconstruction. I would want to see several examples analogous to Fig.4d (although other nanomaterials are also suitable) where the noise in the measurement is smaller than the feature size of the nanoscale sample in order to be convinced that meaningful nanoscale QPI has been accomplished.

If the authors are able to demonstrate unrivalled speed in nanoscale QPI imaging (as they suggest is possible) as well as demonstrate detection sensitivity sufficient to compellingly reconstruct the morphology of nanoscale objects, then I believe this work would then merit publication in this journal as it would clearly demonstrate a strong level of progress in experimental achievement that would genuinely inspire and interest fellow researchers and influence new avenues of experimental investigation.

Aside from these criticisms, I have several smaller questions that I would like addressed:

1) In the introduction the authors write that “QPI of weak scatterers remains complex to achieve”. I would like to see a slight development of this statement. Why is it complex? What is difficult?

- 2) What is the stability in the phase-shift one can achieve in the reference beam via the PM-SLM
- 3) Upon heating the PM-SLM, are there thermal distortions in the device? Do these affect imaging?
- 4) Does one get any back-reflections off the PT-SLM, and does this affect the imaging of the sample?
- 5) Since Gold nanorods have a strong polarization response in their illumination and scattering, how are the authors sure the gold nanorod layer – which lies in the optical path – does not introduce any polarization effects as claimed?
- 6) Can the authors explain the spatially inhomogeneous shifting (distortion) demonstrated in Fig.2(b-e)? It is to their credit that this issue is discussed clearly and such a characterization is shown, but I feel it requires an explanation. Also, I do not agree that such a spatial shift is acceptable if one is wanting to localize and track the nanoparticles. You can localize your particles to a precision much smaller than the drift you get, and since the drift is non-uniform you cannot easily account for it.
- 7) The model which is fitted in Fig.3d appears to not follow your data very well. Your data nicely is self-consistent, but clearly not linear. Is this model suitable to be used here? Also, the figure has no y-label. As a minor detail, I would like a better description of what 'Focus position' means in this context. Is the sample moved closer to the objective or moved away? I assume 0um denotes being in focus.
- 8) As mentioned earlier, I think the manuscript would benefit from a more explicit discussion and characterization of the detection sensitivity, namely, the precision that you are able to achieve with the z height assignment, with the discussion including the relevant noise sources (PT-SLM, optical noise, vibrations etc).

In summary, I consider this work to be an inventive solution to a novel nanoscale QPI problem and commend the authors for their excellent, thorough and exciting work. What is demonstrated in the manuscript holds genuine promise to have appreciable impact on the field, but the progress presented currently falls a little short of the prestigious level which I believe is required for publication in Nature Communications. I hope the authors are able to compellingly address my concerns as I would very much like to see this work achieve the prominence it deserves and wish them every success in doing so.

We appreciate the time and thorough analysis of our work by all three reviewers. We address all their comments in detail below and provide all requested updates and clarification as well as a significant amount of new data in the attached manuscript and supplementary materials.

Reviewers' comments:

Reviewer #1 (Remarks to the Author):

This paper describes the use of a novel photothermal-based phase-modulation That exploits laser heating of an optical element constructed from the layer-by-layer deposition of gold nanorods. This device is used to impart phase control for quantitative phase imaging; demonstrated here via interferometric scattering microscopy of microtubules.

Although photothermal modulation of refractive index has previously been reported (e.g. Heber ACS Nano 2014, 8, 2, 1893–1898), this paper provides a route to spatial control of the phase profile. The authors also report an impressive response time in this proof-of-concept apparatus of 70 us, and estimate a theoretical minimum of 430 ns.

We thank the referee for the acknowledgment of our substantial progress in the technology. We also added a citation to the paper by Heber et al. in the introduction (ref 33) and put it into the context of our work "The photothermal control enabled all-optical light modulation based on local tuning of the birefringence of liquid crystals,³³...".

-The potential advantages of this method are clear, there remain a number of technical hurdles that have not been overcome in these current experiments, and do limit its utility:

We agree with the referee that the technology of photothermal phase modulation opens up numerous technical hurdles that need to be addressed. In the revised manuscript we considerably expanded the range of characterizations to address the reviewer's concerns, as outlined below.

(1) The spatial light modulation demonstrated by the authors is of a fixed circular phase mask, and is not used as an active optical element.

The scope of our manuscript is in implementing PT-SLM in a new type of QPI system and we focus on providing exhaustive characteristics of the performance parameters related to this technique. In the expanded introduction section, we have highlight previous studies (ref 30, 31, 32) that have investigated the technology as a general active optical element "...shaping temperature profiles by nanoparticle distribution³⁰ or spatial modulation of a light wave.^{31,32}". Our results mostly focus on the current advancement. Additionally, we expanded the performance characteristics by characterizing the thermo-induced phase pattern of the optical wavefront using our method, which is provided in a new supplementary section (S5).

(2) The authors demonstrate a 70 us switching time upon heating of the phase element. However, to be really useful, it is fast switching on and off that is required for high speed phase imaging. Cooling takes longer, of the order of 2 ms in the current device.

We agree that the presentation in Figure 2 was unclear when designing our experiment to show the optimized response time only from off to on. To clarify the general design, we provided new data and showed an advanced heating profile resulting in the temporal response to the heating and cooling cycle, both, within the 70 μ s switching time, Figure 2i. Furthermore, we added a discussion on the limitation of the rates of the heating and cooling profile in Figure 2g.

(3) The example application demonstrated by the authors is microtubule iSCAT imaging. It is clear that this is intended as a demonstration of their phase element's capabilities, but worth noting that no specific advance in our understanding of microtubule biology was expected. Thus the impact of the work must be judged on the novelty and importance of this new thermally-driven phase element. For spatially homogenous phase elements, even simple piezo-driven mirrors have resonant frequencies of many 10's of kHz, faster than the overall framerate that this current method could achieve. If the improved microsecond step response time had, for example, been used to measure a physical or biological phenomena its impact would be significantly higher.

We agree that spatially homogeneous phase modulators allow for faster rates of modulation compared to spatially resolved modulators and they are very well suited for two-arm interferometric microscopy used in some configurations of quantitative phase imaging. Indeed, we explored several approaches of phase modulation on our way to the nanoscopic quantitative phase imaging, which resulted in either compromised sensitivity and stability or severe imaging artifacts due to diffraction on available spatial light modulators. The development of a new generation of photothermal spatial-light modulator was directly motivated by this experience. The demonstrated example of the 3D reconstruction of a simple MT network shows a substantial advancement over conventional QPI capabilities and directly challenges a conventional task of, e.g., a 3D reconstruction of a single cell.

To provide more evidence of the breakthrough character of the method, we applied the nanoscopic QPI method for three-dimensional tracking of a single labeled diffusive protein on the surface of a MT illustrated in Figure 5. We followed the trajectory of the particle at a 200kHz rate and every 1 ms, we obtained the full phase information of the scattered light to reconstruct the 3D shape of the trajectory. Interestingly, we show that the molecule undergoes a clear 2D diffusion confined to the MT surface, which is a behavior that cannot be deciphered at a conventional 30 fps video-rate. By this means, we explain how the diffusive MT-associated protein navigates complex MT networks.

There are a number of other more minor considerations that I hope the authors might address. Most pressing is their description of separating the interferometric signals from the two crossing microtubules shown in figures 4 and S7. We are left to guess precisely how these two signals were 'separated', from S7 it looks like x and y components of the original image were analysed separately - relying on the 90 degree crossing in this particular image, but this needs to be made clear.

We thank the referee for this comment. We extended the relevant explanation in the Supplementary section S9 and in the description in the Results to clarify how these two signals were separated: "We averaged the cross-section pattern of each of the microtubules and subtracted these patterns from the original image to obtain the contribution of a single-microtubule as shown in Figure 4b-c (details in Supplementary Information S9).".

Reviewer #2 (Remarks to the Author):

Robert et al. suggest and demonstrate a polarization-insensitive and fast switching SLM, based on the photothermal effect. The approach, in my opinion, could have strong potential in microscopy due to these inherent advantages, and the paper is well written. Disappointingly, the biggest potential advantages of this SLM – insensitivity to polarization, and fast switching - are not demonstrated experimentally. In this sense, this paper presents a limited proof of concept; all the presented experiments could have been performed using existing well-established methods, e.g. a deformable mirror.

We thank the referee for emphasizing the strong potential of the method. To address the request for stronger experimental support of the main features, we added experimental data for the polarization-insensitive response of the SLM and substantially expanded the manuscript with a new experiment of fast 3D tracking of a single diffusive protein on the surface of a MT, as detailed in the response to reviewer #1.

This is without mentioning the limited spatial resolution that is demonstrated (a single disk of 60 um diameter).

We have provided new data and more details on the spatial resolution in the new supplementary section S5 and updated the Results section. The spatial resolution of the thermo-optic effect is fundamentally limited by the heat propagation, as detailed in the manuscript, and the underlying mechanism is dictated by the reciprocal Green's function, which diverges at the origin and decays to infinity. In other words, in a semi-infinite configuration, the heat of a point-source, such as one nanoparticle, can decay on few-nm length scales, suggesting an infinitesimally fine spatial resolution; however, the gradient at the edge of a large heated area depends on the size of the heated area and cannot be simply characterized with a spatial resolution figure. Here, we solved the intrinsic limit of the broad temperature gradient and advanced the control of the phase profile (Figure 1g) for practical use, including the application implemented in our MS.

Specific comments/suggestions:

1. The results in this manuscript would be much stronger if the authors could demonstrate an application that gains from the unique potential of their beautiful photothermal SLM concept. Probably the easiest demonstration to perform would use the fast switching aspect.

We thank the referee for this suggestion. Indeed, we have been able to demonstrate an application by taking advantage of the fast switching, at the same time providing new fundamental insight into the diffusive motion of microtubule-associate proteins.

2. A clearer statement of novelty would help. What exactly are the differences between past photothermal SLMs and the current one?

We thank the referee for this comment. Past photothermal SLMs are limited by the physics of heat propagation, merely creating a gradient lens effect upon inducing the temperature change and the size

of the heated area limits the response time. We overcome these limitations in the new geometry introducing an upper limit on the spatial resolution (illustrated in Figure 1g) and the response time. In the revised manuscript, we added more details about the mechanisms of the SLM technology involving thermo-optic modulation and discussed the novelty. In the Results we add description of the limitations of the conventional approach to thermo-optic SLMs “The usual concept of thermo-optic phase modulation is based on the semi-infinite heat propagation in the thermo-optic material [...] the spatial resolution and the response time is intrinsically limited by the dimension of the thermal-lens.” We follow by comparing the particular advancement in Figure 1g “The green curve (corresponding to the largest thickness of the thermo-optic material) illustrates the phase profile of a thermal lens, which is similar to that generated through a semi-infinite configuration of thermo-optic wavefront modulation.” And we summarize the technical advancement in discussion “The new PT-SLM approach maximizes the phase gradient at the edges of the modulated areas, suppressing the effect of the thermal lens associated with the thermo-optics modulators.”

3. The microtubule experiment raises a few questions:

a. What is the value of “n” used to calculate the z value from the phase? Is n of the microtubule assumed to be equal to n of the environment? In the crossing, for example, there is a microtubule under another one. Is this accounted for? Or assumed negligible?

This is an excellent point to address. In a nutshell, the MT is an extremely small structure having only 25 nm in diameter, which compares to the propagation of light confined to the diffraction-limited spot having one order of magnitude larger diameter. Although the presence of an adjacent MT might have an influence on the evanescent field of the induced dipole that generates the scattered light, the perturbation in the propagating field is only in the order of approximately 2%. Therefore, in the vertical coordinate extraction, we used the refractive index of the environment and we assumed that the near-field coupling of the scatterers was negligible. We have updated the results accordingly: “We used $n = 1.33$ (the refractive index of the surrounding medium) in this calculation. The effect of a microtubule of 25 nm in diameter on the propagating field is on the order of only 2% and the near-field coupling of the two microtubule scatterers is considered negligible.”.

b. The authors report: “vertical displacement at the location of the microtubule crossing peaking at 40 nm with a full width of 500 nm”. This number, 500 nm is even validated using AFM on a similar (?) crossing. But what exactly is 500 nm wide? The microtubule (unlikely)? It is not clear.

We agree that this parameter needs further clarification. The length of 500 nm refers to the length of the transition region of the vertical displacement of the MT measured along the axis of the MT. Conversely, in the direction perpendicular to the MT the optical image is diffraction-limited whereas the AFM scan is limited by the geometry of the AFM tip. We modified the relevant section accordingly: “The displacement peaks at 40 nm and corresponds to a transition region along the axis of the microtubule with a full width of 500 nm which is larger than the microscope point-spread function.”

4. I am confused by the following: the text states: “The temporal heat profile begins with a 100- μ s pulse with a heating laser power of 110 mW. It is followed by a sharp decrease in the heating laser power (44

mW) to stabilize the temperature increase towards a steady state.” But from fig. 2h it seems the HLP goes to 0.

Indeed, there was a typo in the manuscript and we apologize for the confusion. This sentence refers to Figure 2i. We have corrected the relevant text.

5. Figure 3d: which phase is plotted? In the center of the nanoparticle?

Correct. We plot the phase as measured in the center of each nanoparticle, where the contrast is the highest. We added the clarification to the manuscript:” A plot of the variation of the phase at the centers of the two images of nanoparticles as a function of the focus displacement over 1.5 μm is shown Figure 3d.”.

6. Figure S8a – how long is the scalebar?

We have updated the figure with the length of the scalebar (500 nm).

7. A major application that uses SLMs and would benefit from the proposed concept is fluorescence-based 3D single molecule localization microscopy and particle tracking. This could be included in the introduction as motivation along with relevant references.

We thank the referee for this suggestion. Indeed, point-spread-function engineering is one of a number of very exciting applications of the SLM technology. We have expanded the introduction accordingly: ”These tools have found numerous groundbreaking applications such as point-spread-function engineering for 3D fluorescence localization,¹⁹ imaging through scattering media,²⁰ and stimulated emission depletion (STED) microscopy.²¹”

Reviewer #3 (Remarks to the Author):

Robert et al give a proof-of-concept demonstration of a novel photothermal spatial light modulator to introduce a phase shift between scattered and reference beams in a common-path iSCAT microscope. This permits them to perform quantitative phase imaging (QPI) on matter such as gold nanocolloids and microtubules, and on the latter the phase information is used for 3D reconstruction of the scatterer.

The paper overall is well-written and structured in a clear manner. To their credit, the authors provide plenty of quantitative experimental details and thorough characterization of the photothermal spatial light modulator (PT-SLM). The authors also provide a wealth of supplementary information which helps support the claims in the main manuscript, and it is clear a great deal of care and attention has been paid in the creation of this work as well as the production of the manuscript. The paper is a pleasure to read.

Thank you.

The combination of fast QPI with iSCAT is an interesting idea. While neither QPI or interference microscopy of nanomaterials is new, their successful combination provides fertile ground in what I believe would be of interest and practical use within communities such as nano-optics and microscopy, as well as single molecule/particle (bio)imaging.

We are happy to read that the essence of the work stands clearly above the current state of the art and holds clear potential in microscopy applications.

My main concern, however, is that as the manuscript stands, I do not feel the authors have demonstrated a sufficient level of progress that would convincingly enable themselves new lines of scientific inquiry (e.g. nanometric QPI of uncharacterized or dynamic samples) or motivate interest and influence other in the field to adopt the method presented. This is not to take away from the accomplishments of the authors, indeed the work presented is quite impressive and the thorough level of care taken leads to one to be convinced of the exciting possibilities of QPI-iSCAT.

My reservation with endorsing the manuscript is that I do not see demonstrated enough advance in progress over what else exists in the literature that is warranted in such a high-impact journal or expected by the community to open up real lines of new experimental inquiry.

We took this comment very seriously and thank the reviewer for the feedback. As we have already responded to the comments made by the other referees, we are happy to confirm that we have successfully implemented the ultrasensitive QPI-iSCAT technique with millisecond temporal resolution in an ultrafast tracking experiment. The presented experiment illustrates the main advantages of the PT-SLM and provides a new and unique insight into the interaction of single proteins with the surface of a microtubule. This is the first ultrafast fully quantitative 3D tracking experiment, which we believe has a strong potential to become a cornerstone for future studies of mapping single-protein interactions at the nanoscale.

I will outline my specific main concerns below and follow with some minor questions.

The device demonstrated (PT-SLM) is smart and inventive solution to introducing a phase-shift for a common-path interferometer. The extensive modeling and characterization of the device by the authors is a strong point, and the performance claimed by the authors is convincing. However, the device by itself does not merit publication within such a journal since it does not consist of any new discoveries, only a synthesis of existing materials and technologies, albeit in a novel and clever way to tackle an interesting and useful problem.

We appreciate the strong points of our work outlined by the reviewer. Indeed, the aim of our work was to introduce a new generation of QPI techniques by breaking the limits of conventional microscopy and bringing the beautiful and widely applicable principles of QPI into the performance level of nano-optics. We agree that we did not discover any new physics, however, the technological leap is substantial because it is motivated by experiments not available to date. iSCAT is often marketed as a simple and most straightforward route to imaging ultraweak scatterers including single proteins. However, the sensitivity does not come for free and requires reducing the number of optical elements to the necessary minimum required for optical imaging where any additional scattering, diffraction, or phase

instability severely hamper the sensitivity. Therefore, until today, it was never possible to combine iSCAT technology with an existing SLM device to simply synthesize a quantitative phase image of ultraweak scatterers.

To further support the novelty of our work, we have expanded the performance characteristics and have provided new data demonstrating experiments that would not be possible without QPI-iSCAT.

Since the authors introduce the device for the application of QPI “at the nanoscale”, this claim requires scrutiny.

The authors claim the PT-SLM can operate quickly and also permit QPI with exceptional nanoscopic sensitivity. These two important claims need to be more strongly and convincingly evidenced in practice for the work to be justifiably considered to advance the progress of the field.

Firstly, the speed by which the full phase can be swept, and hence sampled, is exciting, and hence would be something the community would want to see – and use! I would like to see a demonstration of a time-dynamic nanoscale sample being imaged fast with full phase information. For example, Kandel et al (DOI: 10.1021/acsnano.6b06945) perform SLM-QPI microscopy to image microtubule mobility at video frame rates. I do not see the authors demonstrating an improvement in imaging performance (speed and sensitivity) beyond what was achieved in that work.

Based on the comments made by all three reviewers, we understand that the application of the speed is an essential step that the community wants to see. In this revised version of the paper, we implemented the measurement concept into a real-time quantitative phase detection device and present data of the imaging of a time-dynamic nanoscale sample at high-speed with full phase information extracted at a sub-millisecond sweep. Similar to the work by Kandel et al., we focused on the dynamics of a MT-associated protein, however, the family of diffusive proteins we used is known for its significantly faster and random movement along the microtubule. We show that high-speed tracking is essential for a correct understanding of the protein behavior and we also show that reducing the temporal resolution to a video-rate of 30 fps would result in a major misinterpretation of the interaction.

Secondly, besides how fast one can extract full-phase information, I believe what is also important to this work is the sensitivity one can perform QPI on nanoscaters. This essentially reduces to how well one can determine the phase from such a weak scatterer. This is where I believe iSCAT enters, and I would want to see a more convincing demonstration of the superior sensitivity that iSCAT is capable of (for example, as demonstrated in previous works of the corresponding author) being applied to give “exceptional” nanoscopic sensitivity in QPI and 3D morphology reconstruction. The measurement surrounding Figure 4 is good, and the authors to their credit include AFM measurements in corroboration of their QPI of the microtubule. I accept the authors claim to resolve the crossing of two microtubules, with reconstruction of their height plausibly showing one lying upon another. My concern however is that the reconstructed surface map appears quite noisy, and so I would always have trouble knowing whether the other nanoscale bumps on the tubule are real or noise – hence this is not so practically useful. I appreciate the authors argue the morphology is related to the surface preparation (Supporting Information 8), but no direct evidence is given that their imaging is sufficiently

precise for these features to be real and not simply noise.

We agree that the longitudinal profile of the microtubule 3D image is a performance demonstration rather than a benchmark of the phase sensitivity. We have addressed the concern of this reviewer in several steps. We analyzed the phase stability and modulation reproducibility of the SLM itself (illustrated in Figure 2j,k, and m). Then we characterized the phase extraction reproducibility with a static object as well as the phase-extraction uniformity within the microscope field of view (Figure 3 e,f). We associated the scattering phase uncertainty with the image noise and the residual image distortion due to the phase modulation. Indeed, our data show an outstanding phase homogeneity with a standard deviation of 0.4 nm.

This lack of sensitivity is, I feel, important as the QPI image (Fig.4d) is not compelling by itself, nor is it reliable for nanoscale characterization, and only can be confidently interpreted when supported by a separate measurement such as the AFM image (Fig.4e). For that reason, I do not feel the authors have demonstrated sufficient progress to convince that truly reliable and sensitive nanoscale QPI has been achieved in this work, which I would say is required for publication here. I would want the authors to discuss the issue of detection sensitivity with their normal excellent level of care, detail and attention. They do not discuss the uncertainties of their phase reconstruction, nor discuss how good their imaging could be in regard to nm-sensitivity in the reconstruction. I would want to see several examples analogous to Fig.4d (although other nanomaterials are also suitable) where the noise in the measurement is smaller than the feature size of the nanoscale sample in order to be convinced that meaningful nanoscale QPI has been accomplished.

We thank the referee for pointing out important characteristics, which certainly makes our statement more compelling. As requested, we have expanded the discussion and provided more detailed experimental evidence and analysis in the updated manuscript, as also detailed in the previous paragraph.

If the authors are able to demonstrate unrivalled speed in nanoscale QPI imaging (as they suggest is possible) as well as demonstrate detection sensitivity sufficient to compellingly reconstruct the morphology of nanoscale objects, then I believe this work would then merit publication in this journal as it would clearly demonstrate a strong level of progress in experimental achievement that would genuinely inspire and interest fellow researchers and influence new avenues of experimental investigation.

We thank the reviewer for this summary of his report. Based on all the comments, we have provided new experimental data to demonstrate an unrivalled speed and detection sensitivity in extended detail and filled the gaps of our work to merit publication, as recommended by the reviewer.

Aside from these criticisms, I have several smaller questions that I would like addressed:

1) In the introduction the authors write that “QPI of weak scatterers remains complex to achieve”. I

would like to see a slight development of this statement. Why is it complex? What is difficult?

We agree that the statement was oversimplified and unclear. We have expanded the introduction and have provided more details: " QPI has already reached the sensitivity to image the displacement of nanostructures at video-rates and specific applications in plasmonic imaging have allowed characterization of nanoscopic patterns. However higher sensitivity and speed of QPI remain limited by the performance of the available spatial light modulation techniques".

2) What is the stability in the phase-shift one can achieve in the reference beam via the PM-SLM

We have provided experimental data and a new discussion on this topic (illustrated Figure 2j and m). It is directly linked to the phase extraction precision, as discussed in our earlier response.

3) Upon heating the PM-SLM, are there thermal distortions in the device? Do these affect imaging?

This is an excellent question. It is very likely that if there was no thermally induced refractive index gradient, the thermal expansion effect would be measurable. Our estimation indicates a small bend of ~ 30 arcsec can be induced by the heat on the glass coverslip resulting in an effective focal length of the PM-SLM in the order of 20 m when heated. This heat distortion effect, regardless of how marginally it appears, together with any asymmetry of the point-spread function, may explain the very small image distortion characterized in the manuscript. We have added a relevant discussion to the manuscript: "We attribute the marginal image distortion to two effects (i) a possible asymmetry of the point-spread-function having a different effect on the peak and dip localization and (ii) possible thermal distortion of the glass substrate owing to inhomogeneous heating of its surface. Our estimation indicates a possible small bent of ~ 30 arcsec owing to the thermal expansion."

4) Does one get any back-reflections off the PT-SLM, and does this affect the imaging of the sample?

Back-reflections can be spatially separated, resulting in no effect on the imaging. In real experiments with the PT-SLM placed in the Fourier plane, we never experienced any issue with ghost images due to multiple reflections on the PT-SLM piece, neither on the glycerol layer, which is relatively well index-matched with the surrounding substrates. Considering that the scattered field and the reference field are spatially separated at the position of the PT-SLM element, any multiple reflections can contribute only linearly to the resulting image, yielding less than 0.3% change relative to the detected contrast of the sample (4% reflectivity of the glass surface and 7% reflectivity of the sapphire surface).

5) Since Gold nanorods have a strong polarization response in their illumination and scattering, how are the authors sure the gold nanorod layer – which lies in the optical path – does not introduce any polarization effects as claimed?

The gold nanorods are randomly oriented and very dense on the surface and therefore their polarization effect is averaged within a diffraction-limited spot. Furthermore, their interaction with the imaging light is off-resonance and, thus, of small anisotropy. The phase change is generated in the heated liquid thermo-optic medium and therefore isotropic. To support this claim, we carried out a polarization-resolved experiment alongside the newly added stability and reproducibility of the PT-SLM device (illustrated in Figure 2l and m).

6) Can the authors explain the spatially inhomogeneous shifting (distortion) demonstrated in Fig.2(b-e)? It is to their credit that this issue is discussed clearly and such a characterization is shown, but I feel it requires an explanation. Also, I do not agree that such a spatial shift is acceptable if one is wanting to localize and track the nanoparticles. You can localize your particles to a precision much smaller than the drift you get, and since the drift is non-uniform you cannot easily account for it.

We appreciate that the reviewer brought this discussion to our attention. It boils down to the interpretation of parameters of accuracy vs. precision. The purpose of the distortion characterization is to demonstrate that the scattered beam is affected by the PT-SLM modulation in a very small fraction of $\sim 0.5\%$, which agrees with the beam overlap discussed in the paper. If the particle travels 10 nanometers, the apparent displacement will be 10 nm both in Figure 2b and Figure 2d, and will be resolvable by a conventional localization algorithm. Thus, the precision or resolution of the localization is not compromised in nano-QPI. However, if the particle displaces half a wavelength vertically and we compensate for the resulting phase-shift using the inline PT-SLM, the accuracy of the endpoint displacement might be affected by the 10-nm distortion. No other optical localization method provides the accuracy parameter and therefore it is difficult to judge if this performance is good or bad, but we can at least put a number on it. In another example, if the particle hovers at a particular height above the surface and we sweep the phase to measure its height, the phase reconstruction might be limited by the slight displacement or distortion of the PSF. Therefore, this comes back to the discussion of the phase reconstruction sensitivity and phase reproducibility and we address this separately in the discussion. Furthermore, we characterized the stability of the position of the particle localized in the amplitude image to be better than 0.1 nm in the supplementary information S7.

7) The model which is fitted in Fig.3d appears to not follow your data very well. Your data nicely is self-consistent, but clearly not linear. Is this model suitable to be used here? Also, the figure has no y-label. As a minor detail, I would like a better description of what 'Focus position' means in this context. Is the sample moved closer to the objective or moved away? I assume 0um denotes being in focus.

The literature offers two models of the Gouy phase profile of a point source in the focus of a microscope. The simple model is based on a gaussian beam approximation resulting in an arcus tangent profile of the phase. A more elaborate model (ref 42) is based on a Fourier transformation of a disc-shaped back aperture of the microscope objective resulting in a linear Gouy phase profile. Our data matched closely with the theoretical slope of this model, however, features a curvature that we attribute to a nonuniformity in the field distribution at the objective back aperture. We updated the discussion to clarify this mismatch: "The phase profile corresponds to the Gouy phase of a focused coherent optical beam, which is known to follow an arctangent function for a Gaussian beam; however,

in the image of an optical microscope with a circular aperture, the theory suggests a linear dependence on the focus position [...] The residual deviation of the experimental curve from the linear theoretical model indicated more specific field distribution at the microscope objective aperture.”. In addition, we provided further information in the caption of the Figure 3d with a better description of ‘Focus position’.

8) As mentioned earlier, I think the manuscript would benefit from a more explicit discussion and characterization of the detection sensitivity, namely, the precision that you are able to achieve with the z height assignment, with the discussion including the relevant noise sources (PT-SLM, optical noise, vibrations etc).

The manuscript was considerably expanded and also includes further explicit discussion of the detection sensitivity.

In summary, I consider this work to be an inventive solution to a novel nanoscale QPI problem and commend the authors for their excellent, thorough and exciting work. What is demonstrated in the manuscript holds genuine promise to have appreciable impact on the field, but the progress presented currently falls a little short of the prestigious level which I believe is required for publication in Nature Communications. I hope the authors are able to compellingly address my concerns as I would very much like to see this work achieve the prominence it deserves and wish them every success in doing so.

We thank the reviewer for the conclusion of his report.

In summary of our response to the reviewer’s comments, we provide the demonstration of phase-detection sensitivity and unrivaled speed in nanoscale QPI and experimentally show that without the performance achieved the data would suffer from significant time-averaged bias. Together with numerous improvements of the manuscript and technology characteristics provided, we believe that our work merits publication in Nature Communications as it clearly demonstrates a strong level of progress in experimental achievements that will inspire and influence new avenues of experimental investigation.

REVIEWER COMMENTS

Reviewer #1 (Remarks to the Author):

The authors provide improved evidence of the temporal resolution of the technique, sufficient to address my major criticisms of the previous draft. Other minor points have also been addressed.

Reviewer #2 (Remarks to the Author):

Robert et al. improved the manuscript, most importantly by including a single particle tracking experiment that utilizes the fast switching capability of their PT device. This is indeed a good experiment for proving this point. I only have one small issue about it given the results:

1. The authors state: "The envelope of the trajectory clearly follows a segment of a cylindrical surface of approximately 40 nm radius (indicated in Figure 5c)". However, looking at figure 5c, I don't see a good fit to a cylinder. It looks pretty planar. So, 'clearly' is not suitable here, I suggest to phrase this more carefully.

Beyond this, I still don't understand some of the answers to my questions, and I am sure that this can be clarified by better explanation in the text:

2. A clearer statement of novelty would help. What exactly are the differences between past photothermal SLMs and the current one?

From the authors' response to my question:

... we summarize the technical advancement in discussion "The new PT-SLM approach maximizes the phase gradient at the edges of the modulated areas, suppressing the effect of the thermal lens associated with the thermo-optics modulators."

This response is not clear to me. My question was exactly this - what IS "the new PT-SLM approach"? For example, is this the first time that the thickness of the 'sandwiched' layer is used as a parameter to tune the resolution? Is this the first implementation that uses a sandwich architecture at all? Etc.

3. The microtubule experiment raises a few questions:

a. What is the value of "n" used to calculate the z value from the phase? Is n of the microtubule assumed to be equal to n of the environment? In the crossing, for example, there is a microtubule under another one. Is this accounted for? Or assumed negligible? We used $n = 1.33$ (the refractive index of the surrounding medium) in this calculation. This is conceptually a bit strange, because if the refractive index of the MT was identical to the surroundings, then it would be invisible.

b. The authors report: "vertical displacement at the location of the microtubule crossing peaking at 40 nm with a full width of 500 nm". This number, 500 nm is even validated using AFM on a similar (?) crossing. But what exactly is 500 nm wide? The microtubule (unlikely)? It is not clear.

The displacement peaks at 40 nm and corresponds to a transition region along the

axis of the microtubule with a full width of 500 nm which is larger than the microscope point-spread function.

I don't understand what a "transition region" is. Transition of what?

Reviewer #3 (Remarks to the Author):

The Authors have carefully responded to specific criticisms raised in the initial review. I am satisfied by their active responses to our initial remarks and also find their corrections to the manuscript concise and appropriate. I find the authors make a convincing and compelling case for the capability and performance of their technique as described. Furthermore, I believe the technique as presented holds great potential for new avenues of label-free quantitative bio-imaging at high spatial and temporal resolution.

I still find, however, that the authors have not demonstrated first steps down such new avenues of investigation, instead they use their impressive technique in exemplary imaging scenarios that are already established.

If such new demonstrations are not a prerequisite for publication, then I endorse the corrected manuscript as it now stands. If, however, claims of novelty and impact are to be evidenced by new measurements, then this I believe is still to be shown.

On this matter, I defer a judgement on suitability for publication to the editor as they are best placed to decide what is expected. I expand my argument below.

My original criticism regarding the novelty of the work stems from the fact that the authors claim a combination of sensitive (nm) and fast (kHz+) quantitative phase imaging but did not fully show this. This combination truly allows for new lines of investigation and imaging that existing techniques (including, but not limited to, other quantitative phase imaging microscopies) cannot offer – hence truly fertile grounds for genuinely novel research with broad appeal. I would like to have seen the authors use their technique to image something no one else has been able to, and hence provide a solution to a problem that has gone unsolved, rather than using their impressive technique to achieve results one can already obtain using existing techniques.

More specifically, the authors show tracking of the mobility of a labeled protein on the microtubule, with their phase-extraction being used to determine the axial position of the particle. This is a nice demonstration of the technical performance of their technique, but one can achieve such performance as already demonstrated in conventional interferometric scattering particle tracking and do so with a simpler experimental set up.

Instead, if the Authors could show, for example, something new such as ultra-high speed precise tracking of motile microtubules on a gliding assay as they collide /intermingle/cross-over etc whilst resolving their nanoscopic 3D topology at high speed, that would be something unique and exclusive to their method. Themselves demonstrating a novel, unique and exclusive measurement afforded by their microscopy alone (or at least the first proof-of-concept type measurements) would undoubtedly satisfy criteria for novelty and high-impact. As it stands, this level of impact and novelty is confidently and robustly implied by the excellent technical characterization of the technique that the Authors demonstrate, but I do not see it has been put into actual practice. I say this because they do not show any first steps along new lines of scientific enquiry that only this new exciting method permits.

As stated above, if the editor judges that such a demonstration is not a necessary criteria publication, then I happily endorse the work as is, else I feel the work would still greatly benefit from such a demonstration.

I would like to end by emphasizing that this criticism results from the high-expectations one places for publication in such high-impact journals. The additional tracking measurements the authors included in the review stage are impressive and certainly praise-worthy, especially given the short time available, and as such I commend the Authors for this genuinely impressive accomplishment. I however feel they do not fully show off the true novelty of the beautiful technique they have developed which is why I continue to wish to see a measurement that goes a little above and beyond what has already been shown in the literature.

We are grateful to all three referees for their time spent with our manuscript, which helped us to considerably improve the publication.

Reviewer #1:

The authors provide improved evidence of the temporal resolution of the technique, sufficient to address my major criticisms of the previous draft. Other minor points have also been addressed.

We thank the reviewer for their time and valuable comments in the first round of revisions. The comments helped us to make significant improvements to the original manuscript.

Reviewer #2:

Robert et al. improved the manuscript, most importantly by including a single particle tracking experiment that utilizes the fast switching capability of their PT device. This is indeed a good experiment for proving this point. I only have one small issue about it given the results:

We thank the reviewer for their time and comments, which helped us improve the manuscript.

1. The authors state: “The envelope of the trajectory clearly follows a segment of a cylindrical surface of approximately 40 nm radius (indicated in Figure 5c)”. However, looking at figure 5c, I don’t see a good fit to a cylinder. It looks pretty planar. So, ‘clearly’ is not suitable here, I suggest to phrase this more carefully.

We agree with the reviewer that the envelope of the data distribution does not offer a clear fit and we thank the reviewer for this note. We clarified the corresponding part of the discussion. In particular, we rephrased this sentence as “The projection perpendicular to the MT in Figure 5c indicates that the highest density of 3D localizations (color-coded in the scatter plot) forms an arc, which we interpreted as a segment of a cylindrical surface with a radius of approximately 40 nm.”

Beyond this, I still don’t understand some of the answers to my questions, and I am sure that this can be clarified by better explanation in the text:

We are happy to provide further clarification as suggested. Please see below for answers to your specific questions.

2. A clearer statement of novelty would help. What exactly are the differences between past photothermal SLMs and the current one? From the authors’ response to my question:

... we summarize the technical advancement in discussion “The new PT-SLM approach maximizes the phase gradient at the edges of the modulated areas, suppressing the effect of the thermal lens associated with the thermo-optics modulators.”

This response is not clear to me. My question was exactly this - what IS “the new PT-SLM approach”? For example, is this the first time that the thickness of the ‘sandwiched’ layer is used as a parameter to tune the resolution? Is this the first implementation that uses a sandwich architecture at all? Etc.

We agree that the novelty of the SLM geometry can be phrased clearer. We updated the discussion to clarify that the novel geometry was optimized by modifying the thermo-optic layer thickness and heat dissipation parameters (the temperature conductive substrate) to tune the resolution and response time:

“The advancement of the PT-SLM resulted from the optimized thin layer of the thermo-optic material and increased rate of heat dissipation, which improved the spatial resolution and modulation speed and suppressed the effect of the thermal lens at the edges of the heated area.” This statement complements the introduction in which we summarized that previous works simply accepted the fact, that heat propagates in spheres and forms thermal lenses. This formulation clarifies the enabling steps of our technical advancement and obeys the recommendation of Nature Communications Guide for Authors discouraging authors from using definitive statements of novelty.

3. The microtubule experiment raises a few questions:

a. What is the value of “n” used to calculate the z value from the phase? Is n of the microtubule assumed to be equal to n of the environment? In the crossing, for example, there is a microtubule under another one. Is this accounted for? Or assumed negligible? We used $n = 1.33$ (the refractive index of the surrounding medium) in this calculation. **This is conceptually a bit strange, because if the refractive index of the MT was identical to the surroundings, then it would be invisible.**

We thank the reviewer for pointing out a possible misunderstanding that we missed. To avoid further misinterpretation, we estimated the error resulting from the refractive index approximation with the surrounding medium and show that it is less than other uncertainties involved in the experiment. We rephrased the sentence as:

“ In the area of microtubule crossing, the light scattered on the microtubule segment further away from the surface passes through the underlying microtubule. Considering the microtubule diameter of 25 nm and refractive index of the microtubule of approximately 1.48,⁴³ we estimated the effective refractive index within the diffraction-limited area of the microtubule crossing to be $n_{\text{eff}} = 1.35$. Therefore, the contribution of the underlying microtubule to the measured phase-shift of the light scattered from the upper microtubule is less than 2%, which is smaller than the measurement uncertainty and, thus, negligible.

b. The authors report: “vertical displacement at the location of the microtubule crossing peaking at 40 nm with a full width of 500 nm”. This number, 500 nm is even validated using AFM on a similar (?) crossing. But what exactly is 500 nm wide? The microtubule (unlikely)? It is not clear.

The displacement peaks at 40 nm and corresponds to a transition region along the axis of the microtubule with a full width of 500 nm which is larger than the microscope point-spread function.

I don't understand what a "transition region" is. Transition of what?

Here we are describing the region in which the distance between the surface and one of the microtubules varies due to the overlapping geometry. We simplified the description as follows:

"The displacement peaked at 40 nm, which corresponds to the maximum elevation of the microtubule in the area where the microtubules overlap. In the direction along the microtubule axis, the profile forms a broad peak with a full width of 500 nm, which is larger than the microscope point-spread function."

Reviewer #3 (Remarks to the Author):

The Authors have carefully responded to specific criticisms raised in the initial review. I am satisfied by their active responses to our initial remarks and also find their corrections to the manuscript concise and appropriate. I find the authors make a convincing and compelling case for the capability and performance of their technique as described. Furthermore, I believe the technique as presented holds great potential for new avenues of label-free quantitative bio-imaging at high spatial and temporal resolution.

We thank the reviewer for acknowledging our first revision.

I still find, however, that the authors have not demonstrated first steps down such new avenues of investigation, instead they use their impressive technique in exemplary imaging scenarios that are already established.

If such new demonstrations are not a prerequisite for publication, then I endorse the corrected manuscript as it now stands. If, however, claims of novelty and impact are to be evidenced by new measurements, then this I believe is still to be shown.

On this matter, I defer a judgement on suitability for publication to the editor as they are best placed to decide what is expected. I expand my argument below.

There are numerous new avenues that the application of the technology can take and some of them certainly offer previously unseen phenomena in biological systems. We presented an experiment that no one else has carried out so far. It is well-possible that there are other methods to study the same system and reach similar conclusions, however, they have not been performed to date. Based on the discussion with the editor, we concluded that although additional experiments would further strengthen our manuscript, they are not strictly necessary for publication.

My original criticism regarding the novelty of the work stems from the fact that the authors claim a combination of sensitive (nm) and fast (kHz+) quantitative phase imaging but did not fully show this. This combination truly allows for new lines of investigation and imaging that existing techniques (including,

but not limited to, other quantitative phase imaging microscopies) cannot offer – hence truly fertile grounds for genuinely novel research with broad appeal. I would like to have seen the authors use their technique to image something no one else has been able to, and hence provide a solution to a problem that has gone unsolved, rather than using their impressive technique to achieve results one can already obtain using existing techniques.

More specifically, the authors show tracking of the mobility of a labeled protein on the microtubule, with their phase-extraction being used to determine the axial position of the particle. This is a nice demonstration of the technical performance of their technique, but one can achieve such performance as already demonstrated in conventional interferometric scattering particle tracking and do so with a simpler experimental set up.

Instead, if the Authors could show, for example, something new such as ultra-high speed precise tracking of motile microtubules on a gliding assay as they collide /intermingle/cross-over etc whilst resolving their nanoscopic 3D topology at high speed, that would be something unique and exclusive to their method. Themselves demonstrating a novel, unique and exclusive measurement afforded by their microscopy alone (or at least the first proof-of-concept type measurements) would undoubtedly satisfy criteria for novelty and high-impact. As it stands, this level of impact and novelty is confidently and robustly implied by the excellent technical characterization of the technique that the Authors demonstrate, but I do not see it has been put into actual practice. I say this because they do not show any first steps along new lines of scientific enquiry that only this new exciting method permits.

As stated above, if the editor judges that such a demonstration is not a necessary criteria publication, then I happily endorse the work as is, else I feel the work would still greatly benefit from such a demonstration.

I would like to end by emphasizing that this criticism results from the high-expectations one places for publication in such high-impact journals. The additional tracking measurements the authors included in the review stage are impressive and certainly praise-worthy, especially given the short time available, and as such I commend the Authors for this genuinely impressive accomplishment. I however feel they do not fully show off the true novelty of the beautiful technique they have developed which is why I continue to wish to see a measurement that goes a little above and beyond what has already been shown in the literature

REVIEWERS' COMMENTS

Reviewer #2 (Remarks to the Author):

The authors addressed my concerns adequately.